

# High-resolution temperature profiling in the Π Chamber: variability of statistical properties of temperature fluctuations

Robert Grosz[1], Kamal Kant Chandrakar[2], Raymond A. Shaw[3], Jesse C. Anderson[3], Will Cantrell[3], and Szymon P. Malinowski[1]

[1]Institute of Geophysics, Faculty of Physics, University of Warsaw, Pasteura 5, 02-293 Warsaw, Poland
[2]Mesoscale & Microscale Meteorology Laboratory, NSF National Center for Atmospheric Research, 3090 Center Green Drive, Boulder, CO 80301, USA
[3]Department of Physics, Michigan Technological University, 1400 Townsend Drive, Houghton, MI 49931, USA

**Correspondence:** Robert Grosz (rgrosz@fuw.edu.pl)

**Abstract.**

This study delves into the small-scale temperature structure of Rayleigh-Bénard convection (RBC) generated in the Π Chamber under three temperature differences (10 K, 15 K, and 20 K) at Rayleigh number Ra $\sim 10^9$ and Prandtl number Pr $\approx 0.7$. We performed high resolution measurements (2 kHz) with the UltraFast Thermometer (UFT) at selected points along the ver-
tical axis to explore to what extent the Π Chamber can be described as an idealized RBC flow. The miniaturized design of the sensor featured with a resistive platinum-coated tungsten wire, 2.5 μm thick and 3 mm long, mounted on a miniature wire probe allowed for undisturbed vertical temperature profiling spanning from 8 cm above the bottom surface to 5 cm below the top surface. The resulting rich dataset comprised both long (19 min) and short (3 min) time series, revealing significant variance and skewness variability in the temperature distributions near both surfaces and in the bulk (central) region linked
with local thermal plume dynamics. We also identified three spectral regimes termed inertial range (slopes of $\sim -7/5$), transition range (slopes of $\sim -3$) and dissipative range, characterized by slopes varying $\sim -7$. Furthermore, the analysis showed a robust relationship between the periodicity of large-scale circulation (LSC) and the temperature gradient, describable by an exponential relation. Notably, the experimental findings demonstrate strong agreement with Direct Numerical Simulations (DNS) conducted under similar thermodynamic conditions, illustrating a rare comparative analysis of this nature.

## 1   Introduction

The convection-cloud chamber, officially named the Π Chamber, represents one of the most advanced facilities for controlled experiments on cloud microphysics (Chang et al., 2016). Its design allows for reproducible and controlled measurements across a wide range of temporal scales, from minutes to days, while maintaining stationary thermodynamic forcing. It operates in two modes. The first mode utilizes pressure reduction to simulate updrafts in the atmosphere. In the second mode, it induces
Rayleigh-Bénard convection, where air in the chamber is heated from below and cooled from above. In the present study we investigate temperature fluctuations in full spectrum of scales in the chamber operating in the second mode. We focus on small-scale temperature fluctuations in a course of turbulent mixing inside the chamber, since the facility is designed for research on



aerosol-cloud interactions in a turbulent environment (Chandrakar et al., 2018a, b; Desai et al., 2018, 2019; Chandrakar et al., 2020; Prabhakaran et al., 2020; MacMillan et al., 2022). Unlike typical RBC experiments, the chamber includes side windows

and various mounting points for microphysical instrumentation, which introduce asymmetries between the upper and lower plates. Thus the detailed (e.g. thermal) characterization is required to evaluate how closely the flow resembles classic RBC flows. It is important to note that this study does not aim to extend beyond conventional RBC research, which often involves day-long averaging. Nonetheless, we report very rich statistics compared to previous highly-resolved measurements within the thermal boundary layer of RBC (du Puits et al., 2013; du Puits, 2022; du Puits, 2024).

Our work primarily focuses on understanding the small-scale and short-term variability of thermal conditions within the facility, emphasizing the importance of absolute temperature. This aspect is crucial for more comprehensive studies on aerosol interactions with water vapor and droplet growth/evaporation in a turbulent environment. However, a few selected results are presented in a non-dimensional form (see Appendix B). Worth highlighting paper is terms of small-scale variability is the recent work on the subgrid scale scalar variance modeled in large eddy simulations for Ra $\sim 10^8$–$10^9$ (Salesky et al.,

2024). Our approach was to collect high resolution (2 kHz) temperature time series using the UltraFast Thermometer (UFT) at selected locations in a vertical profile near the axis of the chamber and to perform statistical and spectral analysis investigating small-scale structure of RBC under laboratory conditions.

UltraFast Thermometers (UFTs) have been specifically designed for airborne in-cloud measurements, achieving resolutions down to scales within and below 1 cm, effectively reaching the dissipation range. Successive iterations of the UFT family

(Haman et al., 1997, 2001; Kumala et al., 2013) have utilized similar sensing element–a resistive platinum-coated tungsten wire, 2.5 μm thick and 5 mm long, mounted on a small vane to adapt to local airflow dynamics. In the next sensor versions (Nowak et al., 2018; Siebert et al., 2021), the vane has been removed, leading to further miniaturization of the instrument's dimensions and the implementation of a custom-built electronic system. The current iteration (UFT-2B) has undergone testing i.e. during the recent EUREC[4]A campaign (Stevens et al., 2021). The 3 mm long sensing wire is spanned on an industry-

standard miniature wire probe, allowing for easy exchange of the sensing head (see Fig. 1).

Not only small-scale fluctuations are important but also understanding of changes in the LSC on distributions of mixing ratio, temperature, and supersaturation inside the cell. The established LSC period in the Π Chamber at the temperature difference of 12 K, was approximated to $\tau_{12} \approx 72$ s (moist convection characterized by a mixing ratio of 7.55 g kg$^{-1}$) (Anderson et al., 2021). In this paper we investigate LSC for three temperature differences ($\Delta T$): 10 K, 15 K, and 20 K showing a variability

of periodicity which can be described by the exponential function.

Over the years, numerous attempts have been made to characterize the statistical aspects of thermal convection in convective cells. Comprehensive overviews of recent advancements in natural convection are provided by Fan et al. (2021) and Lohse and Shishkina (2024), along with references therein. A more detailed analyses of statistical properties of the temperature field in RBC has been explored in recent experimental (He et al., 2018; Wang et al., 2019, 2022), theoretical (Shishkina et al., 2017;

Olsthoorn, 2023), and numerical (Xu et al., 2021b) studies where the authors characterized boundary layer and mixing zone of convective flows. Some investigations aimed at describing buoyant thermal plumes departing from the thermal boundary layer, contributing to the overall heat flux through LSC in a wide range of Rayleigh numbers (Ra ranging from $10^7$ to $10^{14}$) (Liu





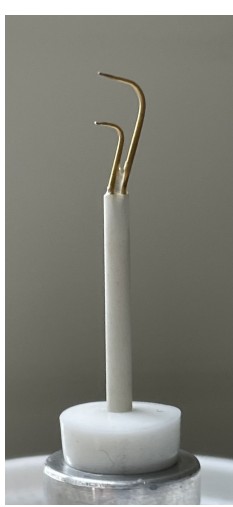

**Figure 1.** UFT-2B head sensor. A parallel to the mean flow, tungsten wire (2.5 μm thick, 3 mm long) spanned on miniature industry-standard wire probe by DANTEC®.

and Ecke, 2011; van der Poel et al., 2015; Zhu et al., 2018; Blass et al., 2021; Reiter et al., 2021; Vishnu et al., 2022; Wang et al., 2022). The studies also examined the effects of cell dimensions, revealing the variable nature of the LSC depending on the cell's aspect ratio ($\Gamma$ = width/height) (Shishkina, 2021). The aspect ratio characterizing the facility (width = 2 m, height = 1 m, $\Gamma$ = 2) corresponds to a single roll with a fixed orientation and pronounced oscillations about the mean position, a result of asymmetries inside the chamber (Anderson et al., 2021) (see Fig. 2). In cases where $\Gamma \gtrsim 4$, a three-dimensional, multi-roll structure has been observed (Bailon-Cuba et al., 2010; Ahlers et al., 2022). Large-scale convective structures have been further explored through DNS, revealing relatively fewer plumes near the sidewalls carrying large heat fluxes, contrasted with more numerous plumes near the cell axis but with weaker heat fluxes, highlighting strong intermittency in this region (Lakkaraju et al., 2012; Chillà and Schumacher, 2012; Stevens et al., 2018; Pandey et al., 2018; Krug et al., 2020; Moller et al., 2021). The simulations also demonstrated the persistence of discrete thermal structures in RBC (Sakievich et al., 2016).

It was recently showed that different regions of RBC can exhibit distinct local dynamics (He and Xia, 2019). As a result, applying a single physical mechanism to the entire convection cell may oversimplify the complex dynamics, as multiple types of force balance can coexist within the same system. On the other hand, studies on the stability of the LSC have proved random reorientations and reversals in both cylindrical setups (Mishra et al., 2011; Wei, 2021; Xu et al., 2021a) and rectangular cells (Vasiliev et al., 2016; Foroozani et al., 2017; Wang et al., 2018; Vishnu et al., 2020), without clearly indicating a superior choice. The validity of Taylor's frozen hypothesis in thermally driven turbulence of RBC, a widely used assumption in the atmospheric community, was examined in numerical simulations by Kumar and Verma (2018). The authors argued that the hypothesis holds true only when a steady LSC is present in the flow. They also cast doubt on the usability of the temperature field for determining whether the Bolgiano-Obukhov (BO, $-7/5$) or Obukhov-Corrsin (OC, $-5/3$) scaling is applicable for turbulent convection. This uncertainty stems from the unclear power-law nature of temperature field spectra and the challenge of contrasting both



scaling factors. Natural convection plays a crucial role in heat and mass transfer in the atmosphere. Despite its fundamental importance, certain characteristics of this phenomenon remain poorly understood and warrant further investigation.

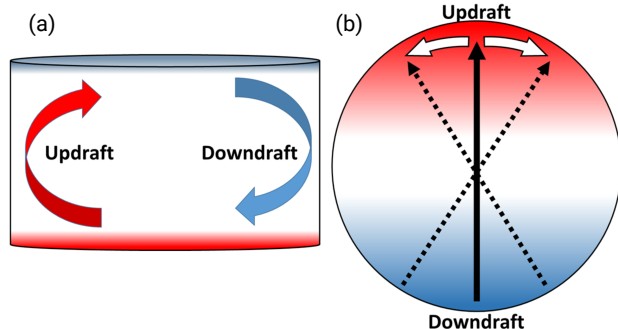

**Figure 2.** Schematic of the cross section of the Π Chamber **(a)** and its plan view **(b)** with the marked LSC. **(a)** The arrows represent the mean direction of the warm updraft (red) and the cool downdraft (blue). **(b)**. The dotted and white arrows show the azimuthal oscillations in the circulation. Figure from Anderson et al. (2021).

From a microphysical perspective, which is the primary application of the chamber, understanding the spatial variability of scalar fluctuations within the chamber, including the properties of the LSC, is crucial. This understanding impacts not only the positioning of instruments inside the chamber but also the strategies for measurements, such as the lengths of measurement time series. Only with a comprehensive understanding of the physics involved can different phenomena be effectively linked together. This is why analyses aimed at addressing the full spectrum of scales are the focus of the present study.

## 2 Methods

### 2.1 Setup and experimental strategy

In our measurements, we utilized the most recent version of the UFT, namely UFT-2B, as outlined in Section 1. The schematic representation of the complete UFT setup can be observed in Fig. 3. The sensor head was affixed to a 1 m probe support and linked to a specially designed 1 mA bridge/amplifier (AMP) using an approximately 1 m standard BNC cable. This amplifier was powered by four AA batteries. Subsequently, the analog signal was acquired by a 16-bit resolution digital-to-analog converter (DAC) from Measurement Computing Corporation (MCC). The DAC had a sampling rate of up to $100 \, \mathrm{kS \, s^{-1}}$ and utilized the dedicated MCC software DAQami. Despite the time constant allowing for about 10 kHz data collection, we opted for an oversampling rate of 20 kHz to facilitate post-processing and filter out artifacts from other lab systems. Using two head sensors during this study, each possessing an approximate resistance of 30 Ω, we attained a UFT sensitivity of approximately $75 \, \mathrm{mV \, K^{-1}}$ after calibrating with a standard thermocouple.

For vertical profiling, the UFT was attached to a 6 mm diameter, 1.5 m long rod with a 90 degree bend at the end. The rod was marked in 3 cm increments to facilitate easy UFT positioning. A sturdy metal stand with two adjustable clamps was used





to secure the rod in a stable, vertical position while allowing for the user to manually move the UFT to the desired location (see Fig. 3). To minimize potential movements of the UFT cabling and sensor head, both were affixed to the rod using simple

adhesive, maintaining the wire in an upward and parallel orientation to the floor.

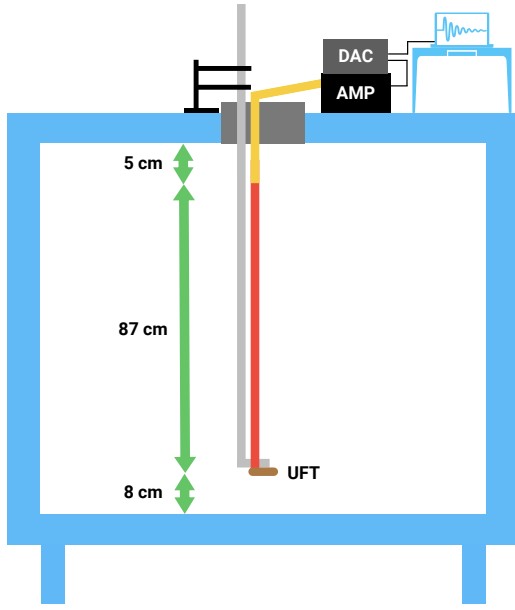

**Figure 3.** Schematic of the setup used during the measurements (diagram not to scale). At the top there was an operations center housing with most of the devices and cabling, including a BNC cable (yellow), an UFT amplifier (AMP), digital-to-analog converter (DAC), and a PC with a DAC software. Inside the Π Chamber, a vertical rod with a curved end (gray) and a UFT sensor (brown) with DANTEC® probe support (red) attached to that end was deployed. The profiling limits were about 8 cm above the bottom and around 5 cm below the top layer. Note that, for clarity, the schematic does not include the cylinder.

We studied the small-scale temperature structure within the convection environment across three temperature differences between the chamber's floor and ceiling: 10 K, 15 K, and 20 K, as detailed in Table 1. The setup included a cylinder, which is not shown in Fig. 3. For a more detailed schematic, please refer to Chang et al. (2016).The Rayleigh number was on the order of $\sim 10^9$ for the set boundary conditions and the chamber height of 1 m. We performed our calculations based on the formula

suggested by Niedermeier et al. (2018), assuming dry convection with an estimated Prandtl number of 0.72.

Our primary focus was on examining the nature of fluctuations throughout the entire vertical dimension, with a particular emphasis on regions near the floor and the ceiling. Consequently, the UFT deployments featured irregular measurement positions. Another consideration was the variable measurement time $t$, ranging from 3 minutes to 19 minutes. This variability stemmed from the LSC period, $\tau_{12} \approx 72$ seconds (Anderson et al., 2021), and the uncertainty surrounding whether different

turbulence properties might be observed for shorter time segments.



A less emphasized aspect was the surface topography. One configuration involved the presence of rough boundaries, consisting of aluminum bars (4 cm wide and 1.4 cm high) positioned on the floor and ceiling forming longitudinal stripes separated by 17 cm intervals. The bars themselves were at a slightly different temperature compared to the rest of both panels (approximately 0.4 K). Subsequent UFT deployments were conducted after removing the bars, aiming to compare temperature
fluctuation properties between the two cases. Unfortunately, a portion of the dataset is invalid due to high battery drainage, resulting in coverage of only one rough boundary case in this study.

As the surfaces inside the chamber reached steady temperatures (refer to Tab. 1), the UFT sensor was initially positioned 8 cm above the floor, near the axis of the cell. Due to the rod's length inside the chamber corresponding to its height, we had to wait for some time to allow the vibrations of the head sensor to dampen. This was really important after each position ($h$)
change but played a crucial role especially in profiling the lower half of the measurement volume. The chamber's flange was covered with a thick foam layer, effectively reducing most mixing events near the opening. Although not an ideal solution, it seemed the most reasonable choice considering the ease of checking the UFT position, as well as the insulating and damping properties of the foam (when coating the rod).

**Table 1.** List of experiments together with the corresponding Π Chamber and UFT settings, and the Rayleigh numbers. Symbols explanation: $T_F$, $T_C$, $T_W$ represent floor, ceiling, and walls temperature respectively, $t$ stands for the measurement time at a given height $h$ above the floor. Names of the experiments are made as follows: type of the measurement ("V" for vertical), $\Delta T$, type of boundaries ($S$ for smooth or $R$ for rough), and time spent at a single position ($L$ for 19 min or no marking for 3 min).

| Experiment | rough/smooth boundaries | $T_F$ [°C / K] | $T_C$ [°C / K] | $T_W$ [°C / K] | $h$ [cm] | $t$ [ min] | Ra [×10$^9$] |
|---|---|---|---|---|---|---|---|
| V10-S-L | smooth | 25 / 298 | 15 / 288 | 20 / 293 | irregular | 19 | 1.1 |
| V10-S | smooth | 25 / 298 | 15 / 288 | 20 / 293 | 8–95 | 3 | 1.1 |
| V15-S-L | smooth | 27.5 / 300.5 | 12.5 / 285.5 | 20 / 293 | irregular | 19 | 1.6 |
| V15-S | smooth | 27.5 / 300.5 | 12.5 / 285.5 | 20 / 293 | 8–95 | 3 | 1.6 |
| V20-S-L | smooth | 30 / 303 | 10 / 283 | 20 / 293 | irregular | 19 | 2.1 |
| V20-S | smooth | 30 / 303 | 10 / 283 | 20 / 293 | 8–95 | 3 | 2.1 |
| V20-R | rough | 30 / 303 | 10 / 283 | 20 / 293 | 8–95 | 3 | 2.1 |

The irregular positions are: 8, 14, 26, 35, 50, 65, 74, 86, 95 [cm].

After completing the measurements, the dataset underwent several basic preparations. These included the removal of electronic artifacts, signal despiking, Butterworth filtering (10th order, 2 kHz cuttoff frequency), 2 kHz averaging, and the trans-
lation of values from voltage to temperature units. Additionally, each time series was consequently normalized by subtracting mean temperature value in the given position (see e.g., Fig. 4).





## 2.2 DNS methodology

Cloud Model 1 (CM1) (Bryan and Fritsch, 2002) in DNS configuration is used for these simulations. The model and setup
are described in detail in Chandrakar et al. (2022, 2023). The computational domain size for DNS is $960 \times 960 \times 500$ grid
cells with a homogeneously 2.083 mm grid spacing in horizontal and stretched grid in vertical (finer near the top and bottom
boundaries). CM1 solves the conservation equation set with the Boussinesq approximation and a prognostic pressure equation
using a three-step Runge–Kutta time integration method with a fifth-order advection scheme. The Klemp-Wilhelmson time-
split steps are used for the acoustic terms in the compressible solver. The time integration of the governing equations uses
an adaptive time step with a maximum Courant–Friedrichs–Lewy (CFL) number of 0.8. A no-slip boundary condition for all
walls is applied, and the temperature boundary conditions (constant temperatures) are the same as the experimental setup. The
simulations use molecular viscosity and thermal diffusivity values at the mean temperature (Prandtl number = 0.72). DNS is
performed for the three experimental cases, V20-S, V15-S, and V10-S, listed in Tab. 1. Outputs from a steady-state period after
the initial spin-up are used for the analysis. Consistent with the experiments, the Eulerian temperature time series are outputted
at 0.0012–0.0015 s intervals from a region near the center of the domain (95–105 cm from sidewalls) at multiple heights from
the bottom surface.

## 3 Results

### 3.1 Deteremination of basic characteristics of temperature profile

The top panel of Fig. 4 provides a sample of temperature fluctuations $T'$ ($T' = T_h - \overline{T}_h$, where $T_h$ represents the temperature
series at a given height $h$, and overline denotes the mean) from the vertical scan of the measurement volume near the axis of
the chamber. The skewed fluctuations observed in the closest proximity to the plates serve as expected temperature evidence of
RBC in the cell. We can observe a smooth transition involving gradual suppression of fluctuations or rather gradual decrease
in occurring thermal plumes as the sensor moved towards the mid-height plane. The reverse symmetry is present in the upper
half of the cell. The nature of these fluctuations aligns with the numerical results of heat fluxes in the bulk region obtained
by Lakkaraju et al. (2012), temperature time series reported in He and Xia (2019) and Wang et al. (2022), and experimental
data provided by Anderson et al. (2021). However, it is noteworthy that all these works primarily focused on specific regions
of the cells, lacking a more detailed insight into the temperature characteristics, especially considering the limited temporal
resolution of the used instrumentation.

The most substantial temperature fluctuations are observed near the floor region. In cases with a flat surface (experiments
summary in Tab. 1), peaks oscillate around 4 K, while rough boundaries scan exhibit fluctuations exceeding 5 K. As the sensor
approaches the mid-high plane, the differences between $\Delta T = 20$ K cases become negligible. Similarly, no distinctions are
apparent near the upper plate, with a maximal amplitude at the level of $-4$ K for both V20-S and V20-R.

In the bottom panel of Fig. 4, two vertical layouts are presented, each illustrating 10 min series near both plates positions
and segregating $T'$ based on the given $\Delta T$. The evident reverse symmetry is notable; however, it is important to highlight



that there are varying amplitudes of fluctuations in each corresponding pair of graphs (same $\Delta T$ but distinct $h$). This variation may result from weaker thermal plumes departing from the top plate, as well as from the not perfectly insulated chamber's flange (mentioned in 2.1), which could lead to minor mixing in the vicinity of the sensor deployment spot. For a more in-depth examination of the temperature fluctuations near both plates, refer to Appendix A.

The temperature fluctuations also manifest oscillations, particularly noticeable in the case of $\Delta T = 20$ K near the plates. However, these oscillations gradually diminish as the temperature difference decreases and as the sensor moves toward the center of the cell. Analyzing V20-S-L at both heights, the periodicity appears irregular but is of a same order of magnitude as observed by Anderson et al. (2021) and therefore corresponds to the LSC. Previous studies have highlighted that the LSC can exhibit various modes around its mean position, leading to phenomena such as out-of-phase oscillations at the top and bottom of the chamber (torsional mode, see Funfschilling et al. (2008)), as well as side-to-side oscillations (sloshing mode, see Xi et al. (2009); Brown and Ahlers (2009)). Cells with very high symmetry might be also characterized by spontaneously cease and reorientation of the LSC to different angular position (Brown and Ahlers, 2009). All these effects are beyond the scope of this investigation but the raw measurements give clear evidence of changing oscillations near both plates.

In Fig. 5, the standard deviation $\sigma_{T'}$ is presented in relation to the sensor position within the cell and illustrates the dependence of the fluctuation level, corresponding to the top panel of Fig. 4. The highest $\sigma_{T'}$ values are observed near both plates, particularly with a clear dominance at the bottom. This asymmetry diminishes as $\Delta T$ decreases, starting with an approximate $0.4$ K disparity in V20-R and concluding with about a $0.1$ K shift in V10-S-L. It's noteworthy that extended measurements yield slightly different values, reflecting a more robust convergence as opposed to $3$ min cases. The bulk region exhibits relatively constant values with comparatively small deviations. Additionally, this region experiences the smallest differences between corresponding $\Delta T$ values. Decreasing $\Delta T$ shifts left $\sigma_{T'}$ values and damps $T'$ in the whole volume. In Fig. B1a we provide non-dimensionalized form of standard deviation.

The surface topography contributes to slightly higher $\sigma_{T'}$ values, primarily in the closest vicinity of the plates. This effect may be attributed to the elevated surface level, potentially leading to varied stages of thermal plume development at the same measurement position. However, these thermal structures are getting mixed with the surroundings, producing approximately equivalent results just a few centimeters higher. As previously mentioned, recent work by He and Xia (2019) emphasizes that each region of the RBC can exhibit its local dynamics, a consequence of overlapping mechanisms that act as drivers for each other. In this specific case, the LSC induces a mixing of all thermal structures originating from the surface, and it can also turbulently propel the thermal plumes due to the irregular topography. The resulting mixing and stronger turbulence in this region might be responsible for the thermal peaks observed in the top panel of Fig. 4.

In Fig. 6, the skewness of $T'$, denoted as $\mu_{T'}$, is analyzed with respect to the vertical positions within the chamber. We use adjusted Fisher–Pearson standardized third moment, expressed as $\mu_{T'} = \frac{N^2}{(N-1)(N-2)} \frac{\overline{T'^3}}{\sigma_T'^3}$, where $N$ represents the number of samples. The findings confirm previous observations, showing positive skewness (associated with warm plumes) near the floor and negative skewness (indicative of cold plumes) just below the ceiling. The third moment is notably influenced by rare events, leading to significant fluctuations in the $3$ min dataset but mostly averaged out in longer segments, resulting in more consistent curves.





(a)

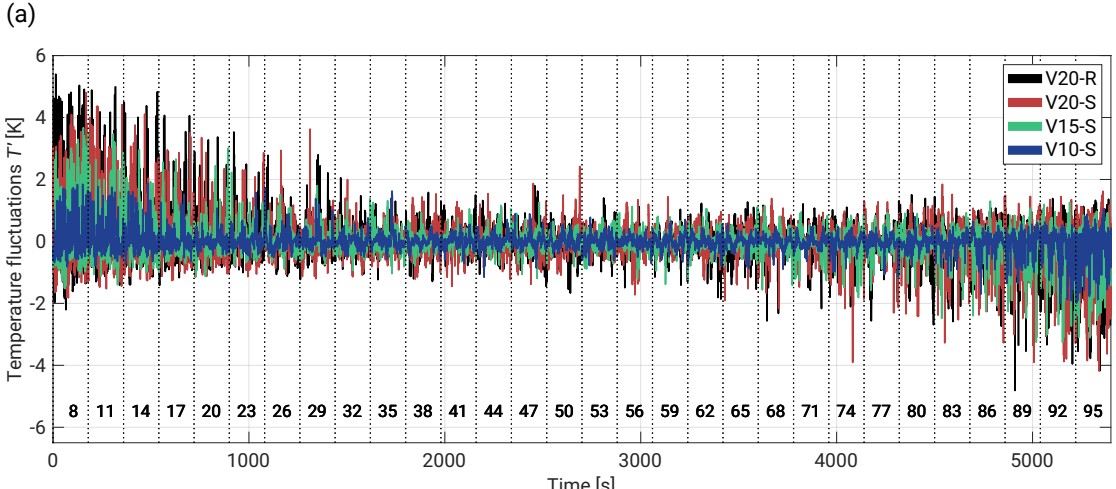

(b)

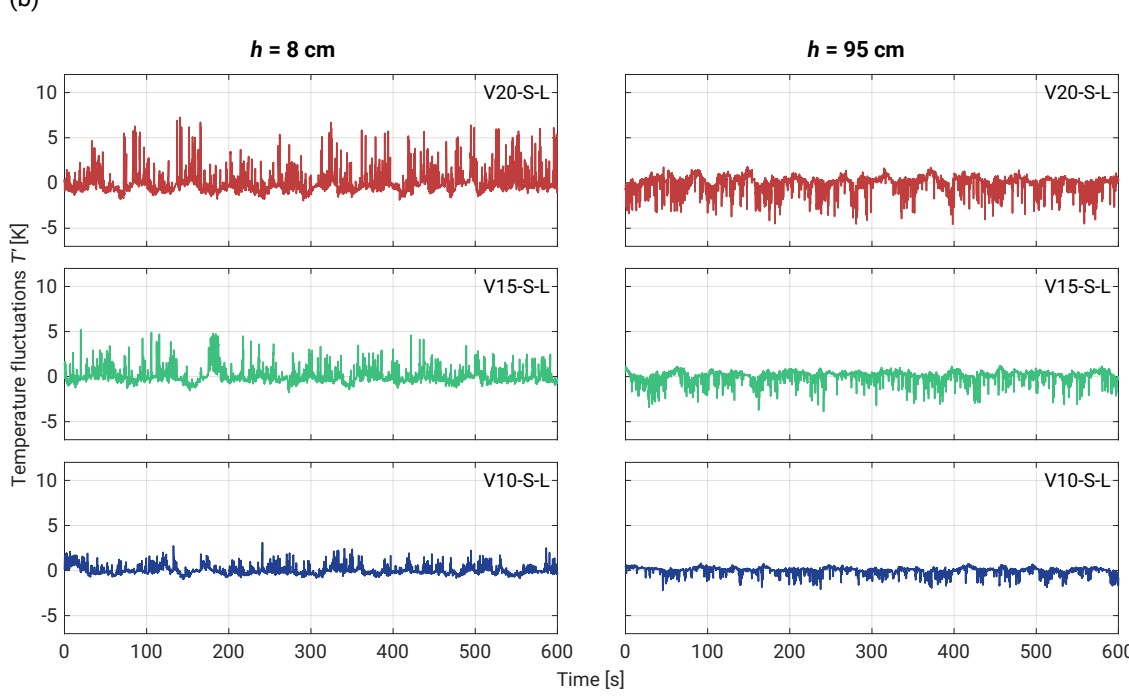

**Figure 4.** Temperature fluctuations $T'$ time series corresponding to different $\Delta T$ that are described in Tab. 1. Top panel **(a)** corresponds to full vertical scan of the cell. The numbers at the bottom denote the position of the UFT in centimeters with respect to the lower plate. The chart includes 3 min series. Bottom layouts **(b)** represents 10 min measurements near the floor ($h = 8\,\mathrm{cm}$), and just below the ceiling ($h = 95\,\mathrm{cm}$, 5 cm below the top plate).



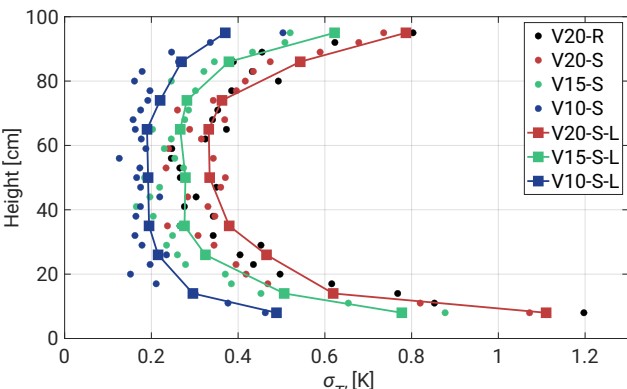

**Figure 5.** Standard deviation $\sigma_{T'}$ with respect to the height of the chamber. Short time series (3 min) are denoted by circles, squares represent longer measurements (19 min). Decreasing $\Delta T$ shifts left $\sigma_{T'}$ values reducing the temperature fluctuations at all positions.

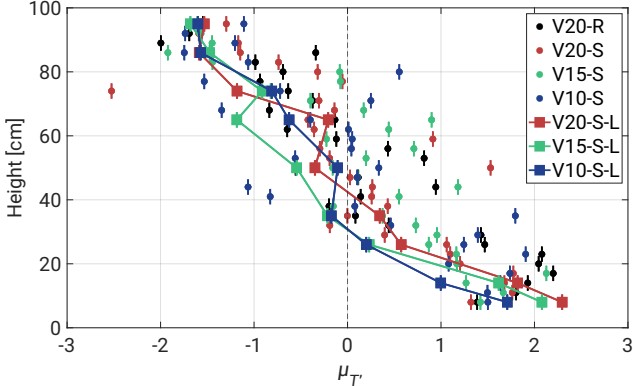

**Figure 6.** Skewness $\mu_{T'}$ with respect to the height of the chamber. Short time series (3 min) are denoted by circles, squares represent longer measurements (19 min). Included uncertainties were calculated using formula $\delta_{\mu_{T'}} = \sqrt{\frac{6N(N-1)}{(N-2)(N+1)(N+3)}}$ where $N$ denotes the number of samples.

The regions near both plates demonstrate a high convergence of $\mu_{T'}$ values with minimal deviations. An interesting observation is noted at the 8 cm region above the floor, where the 3 min records initially exhibit a skewness of about 1.4. This area likely experiences a higher frequency of intense thermal plumes, resulting in a broader range of temperature fluctuations (refer to Fig. 5). As the plume structures develop, $\mu_{T'}$ increases to approximately 2. Then, at the 20 cm level, there is a subtle indication of change in the thermal dynamics of the system. This change may be associated with specific transitions in convective

flow patterns and more intense interaction of thermal plumes with the LSC in the ring layer around the walls and plates (see Fig. 2a). Moving further away from the heated floor, the LSC dominates the existing structures, increasing the dissipation of thermal energy and leading to a decrease in skewness. This results in thermal structures becoming more dispersed, leading to a narrower and less extreme distribution of temperature fluctuations.





However, not all thermal plumes are fully averaged out, especially as the flow around the cell decreases towards more central
regions. This allows some remaining plume structures to reach the central region between 40–70 cm, mixing and resulting in
positive skewness (bottom plumes carry higher energy). This behavior might be also observed in longer records, manifesting
as fluctuations in $\mu_{T'}$ within the 50–70 cm segment. Importantly, the positions of these shifts do not appear to be directly
dependent on $\Delta T$.

A comprehensive understanding of the thermal dynamics requires additional information on the small-scale temperature
field around the axis, its velocity field, and a detailed description of the LSC time evolution.

Upon comparing the topographic effect, we did not observe any major differences and concluded that 3 min records might
be insufficient to investigate the impact caused by the presence of roughness. However, recent numerical work by Zhang et al.
(2018) (for $10^7 \leqslant \mathrm{Ra} \leqslant 10^{11}$ and fixed $\mathrm{Pr} = 0.7$) indicates that there is a critical roughness height $h_c$ below which the presence
of roughness reduces heat transfer in RBC. The authors link this phenomenon with fluid being trapped and accumulated inside
the cavity regions between the rough boundaries. Our approximate calculations for the $\Pi$ Chamber setup indicate the $h_c$ of
approximately 7 mm, compared to the 1.4 cm height of the tiles.

## 3.2   Power Spectral Densities

Power Spectral Density (PSD) of $T'$ was computed using the Welch algorithm. Initial analyses were primarily directed towards
estimating the LSC periods $\tau$ for the given $\Delta T$ and with respect to the measurement position (see Fig. 7a). This involved
utilizing 19 min datasets with window lengths approximately equal to the size of the collected segments. Employing a 50%
overlap between segments and incorporating a high number of discrete Fourier transforms (eight times the window length), we
derived estimates of the LSC periods along with their associated standard deviations. For $\Delta T = 10$ K a modest convergence of
data points is observed, particularly notable within the 60–80 cm region. This resulted in relatively elevated standard deviation
(grey areas denote +/- $1\sigma_\tau$), yielding a period of approximately $\tau_{10} \approx 79$ s. Subsequent $\Delta T$ demonstrated a more uniform
distribution across all levels, accompanied by a gradual reduction in the LSC period to approximately $\tau_{15} \approx 65$ s and $\tau_{20} \approx 57$ s.

The relationship between $\tau$ and $\Delta T$, modeled by exponential function $\tau^\mathrm{e}$, is illustrated in Fig. 7b. The fit exhibits narrow
95% prediction bounds in the fitted region but significantly large bounds outside. The model was constructed using a sparse
dataset consisting of only four data points, including the result obtained by Anderson et al. (2021) at $\Delta T = 12$ K. Consequently,
this limited dataset may not fully capture the true relationship, particularly at lower ($\Delta T < 10$ K) and higher ($\Delta T > 20$ K)
temperature differences. The potential discrepancies could be attributed to a stronger diffusion dominance over convection at
lower $\Delta T$ or more pronounced overlapping thermal plumes at higher temperatures, respectively.

In subsequent PSD analyses, we continued using only 19 min records, as shorter measurements exhibit too much variability
in spectra due to their duration being comparable with the LSC periods. This time-modified window length, approximately
1/9 of the total segment with windows overlapping by half of their length, resulted in 17 individual PSDs that were averaged.
This approach enhances chart readability while maintaining fidelity to the spectral slopes. To collapse the curves representing
measurements from different positions, we followed the scaling method proposed by Zhou and Xia (2001). Fig. 8a plots the





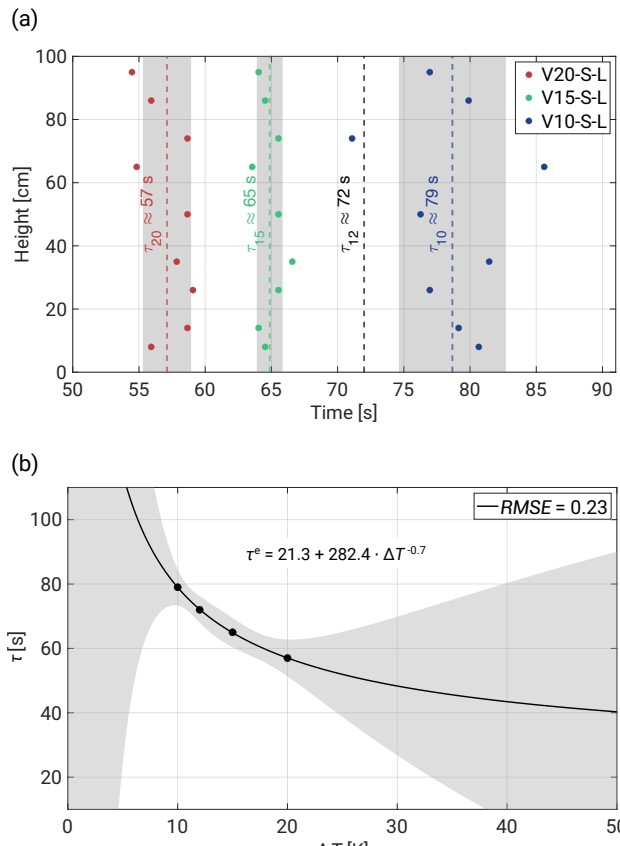

**Figure 7.** Vertical variability of the LSC period with respect to $\Delta T$ **(a)** Grey regions describe +/- $1\sigma_\tau$ and the black dashed line denotes the result obtained by Anderson et al. (2021) for $\Delta T = 12$ K. **(b)** The relationship between $\tau$ and $\Delta T$ modeled by exponential ($\tau^e$) function. The plot includes 95% simultaneous functional bounds, fitted equation, and root mean squared error ($RMSE$).

scaled $f^2 P(f)$ spectrum for the V20-S-L case, enabling determination of the peak frequency $f_{\mathrm{p}}$, around which the PSDs become universal functions. In this case, $f_{\mathrm{p}}$ oscillates around $f = 4$ Hz, exhibiting high convergence across all curves.

In Fig. 8b, we provide a sample of scaled PSD $P(f)/P(f_{\mathrm{p}})$ versus $f/f_{\mathrm{p}}$ in the lower half of the chamber and define three
spectrum regimes. Based on the scaling method proposed by Kumar and Verma (2018) and Zhou and Xia (2001), we conducted also a similar analysis in the wavenumber domain. For more details, please refer to Appendix C. To estimate the slopes, we employed a methodology outlined in Siebert et al. (2006) and Nowak et al. (2021), averaging raw spectra over equidistant logarithmic frequency bins (twenty bins per decade in our case) and fitting using exponential functions. Additionally, to assess the linearity of the slopes in log-log coordinates, we computed the Pearson correlation coefficient $p$ for the resampled points.

In thermally-driven convection flow is not passive but actively driven by temperature (buoyancy) differences. This regime is defined in the literature as the inertial-buoyancy range, with the scalar (temperature) spectrum following BO scaling (Chillá et al., 1993; Ashkenazi and Steinberg, 1999; Zhou and Xia, 2001). Conversely, OC scaling is reported in the inertial-convective




(a)

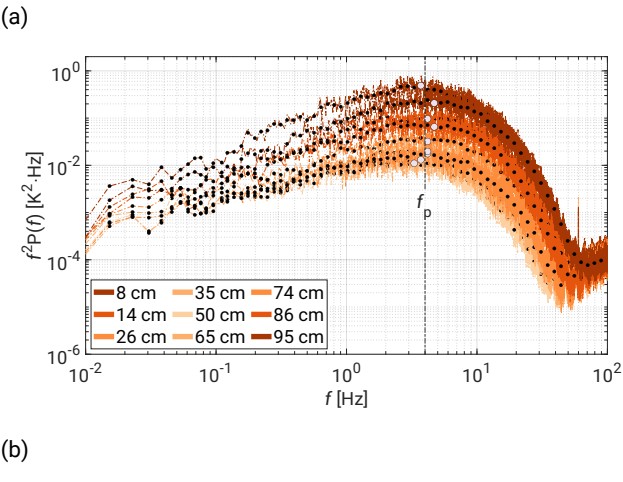

(b)

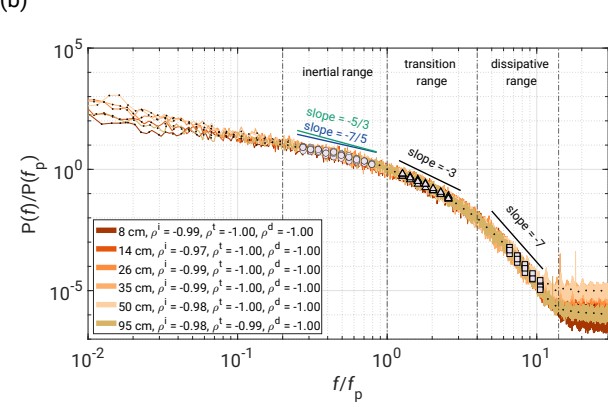

**Figure 8.** Scaled V20-S-L PSD with respect to the UFT positions (color gradients) in the lower half of the chamber (plus top position). **(a)** Scaled spectrum of $f^2 P(f)$ with marked mean $f_\mathrm{p}$ value, $f_\mathrm{p} \approx 4$ Hz. **(b)** PSD $P(f)/P(f_\mathrm{p})$ versus $f/(f_\mathrm{p})$ with three defined regimes: inertial range (circles, $0.2 \leqslant f/f_\mathrm{p} \leqslant 1$), transition range (triangles, $1 \leqslant f/f_\mathrm{p} \leqslant 4$), and dissipative range (squares, $4 \leqslant f/f_\mathrm{p} \leqslant 20$). Each regime is denoted by different markers with an approximate slope value added above curves. The Pearson correlation coefficients $p$ have upper indices to indicate the regimes.

range, where temperature (scalar) is assumed to be passive and has no influence on the flow dynamics (Castaing, 1990; Cioni et al., 1995; He et al., 2014). Interestingly, Niemela et al. (2000) ($10^6 \leqslant \mathrm{Ra} \leqslant 10^7$) and Pawar and Arakeri (2016) (axially

homogeneous buoyancy-driven turbulent flow, $10^4 \leqslant \mathrm{Ra} \leqslant 10^9$) observed both scaling behaviors in their experiments. In the latter study, the authors debated whether the results indicated dual scaling or a gradual steepening of the spectra.

In our case, the regime of $0.2 \leqslant f/f_\mathrm{p} \leqslant 1$, indicated by circles in Fig. 9a, exhibits relatively small variability of PSD slopes. This range was selected based on the highest Pearson correlation coefficient, which limited the fitting region. Our analysis provides no clear answer, as the slopes oscillate between OC and BO scaling, with a slight bias towards $-7/5$. However, as

mentioned earlier, the two slopes are too close to each other to be easily distinguished (see Fig. 8b), leading us to label this regime as the inertial range without its specific profiling.





The regime $1 \leqslant f/f_{\mathrm{P}} \leqslant 4$, indicated by triangles in Fig. 9a, is characterized by slopes oscillating around $-3$, with slightly higher variability for the V10-S-L case. There is no direct reference in the literature for such slopes, and further considerations should explore possible connections between our findings and known explanations.

The first set of explanations follows the rich dynamics of RBC and studies on two- and three-dimensional turbulence. Foundational works in this field include Batchelor (1959) and Kraichnan (1967, 1968), which introduced a specific regime of scales called the viscous-convective range. In this regime, scalar fluctuations are influenced by both turbulent flow and viscous dissipation, but molecular diffusion remains relatively small. In 1967, Kraichnan explored the Batchelor regime in 2D, providing a dual cascade model of inverse energy transfer and direct enstrophy cascade associated with interactions of vortices

and larger coherent structures. The given energy scaling spectra were $-5/3$ and $-3$, respectively. The latter corresponds with the mentioned viscous-convective range, which, in the case of the scalar variance spectrum, should follow $-1$. As described in the overview paper by Alexakis and Biferale (2018), the coexistence of cascades at the same scales is possible, making the outcome spectrum a result of either cascading process. It is also possible for the spectrum to be a superposition of power laws, with each cascade determining the energy spectrum at different ranges of scales. The authors clearly note that the presence

of multiple cascades at the same scales does not imply that both invariants cascade in the same direction. Earlier research by Cencini et al. (2011) showed an intriguing result of the coexistence of both forward and reverse cascades in a situation where energy injection takes place at multiple scales in 2D turbulence.

Based on numerous studies over the years, it is evident that RBC in 3D exhibits very complex dynamics in a wide range of scales, including thermal plumes, vortices, and LSC with their mutual interactions (Fernando and Smith IV, 2001; Xi et al.,

2004; Zhou et al., 2016; Guo et al., 2017; Chen et al., 2018; Pandey et al., 2018; Dabbagh et al., 2020), making it possible for the spectra to be a result of these overlapping processes. Moreover, it seems legitimate to ask to what extent theories based on strong assumptions like isotropy, homogeneity, stationarity, and self-similarity might be sufficient to fully uncover the physical reality, emphasizing the role of anisotropy and non-stationary coherent structures. The real flow might deviate from theoretically predicted spectra scaling, requiring more complex characterization. In the given full spectrum analysis, the nature

of the scale break we observe around $f/f_{\mathrm{P}} \approx 1$ might be linked, e.g., to the dynamic transition between LSC scale dominance, where larger coherent structures form, and the regime of smaller-scale thermal plumes and vortices, which might eventually lead to overlapping power laws and ultimately modifying the final spectra. As an example we present in the Appendix D, an additional analysis showing an estimation of the thermal plumes dominance region close to the chamber center. Based on He and Xia (2019), the logarithmic dependence of $\sigma_{T'}$ on chamber height is demonstrated, which can be interpreted as the balance

of buoyancy and inertial forces.

The second group of possible explanations for the $-3$ scaling regime emphasizes that buoyancy effects in RBC are due to temperature forcing, which drives both large- and small-scale structures. Already mentioned overview by Alexakis and Biferale (2018) points out that making strong assumptions about cascades, including their directions, is not feasible for active scalars. Further arguments highlight the lack of universality between passive and active quantities, particularly in scenarios where there

is a strong relationship between velocity and scalar fields. This leaves open questions about preferential sampling effects of forcing along Lagrangian trajectories of the active scalar field.





The complexity of mentioned problems extends beyond the scope of this paper. Consequently, we classify the $-3$ regime as a transition range between buoyancy-scale processes and molecular dissipative scales.

Based on our high-frequency measurements, we estimated slopes in the dissipative regime to be around $-7$ (see Fig. 9b).
Slope distribution with respect to chamber height is symmetrical, with the highest values near the plates ($\approx -8$) and the smallest in the bulk region ($\approx -7$). There are no direct references in the literature justifying similar slope values. However, similar spectral characteristics can be found in the figures in Niemela et al. (2000) and Zhou and Xia (2001), though without extensive discussion on slopes. There is also a potential connection with the spectral scaling of $-17/3$ in the so-called inertial-diffusive range as proposed in Kraichnan (1968). This continues previous considerations on the complex dynamics in RBC and
the nontrivial scalings observed in many experiments.

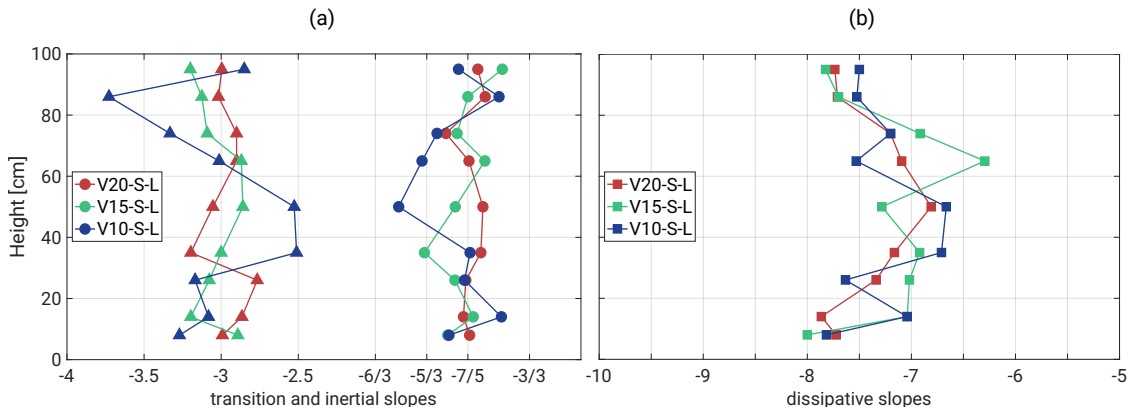

**Figure 9.** Panel of fitted slopes. **(a)** corresponds to transition and inertial ranges respectively whereas **(b)** describes dissipative regime.

Worth noting is also the variability of the noise level (starting around $f/f_\mathrm{p} \approx 12$) with respect to the chamber height, with its highest values linked to the bulk region and the lowest (10 times magnitude difference) representing regions near the plates (see Fig. C1 in Appendix C). This phenomenon is attributed to the mean velocity field and its strong reduction in the central areas of the cell causing the noise to rise.

## 305    3.3    DNS versus experimental data

Another goal of the presented study was to compare the experimental results obtained with the UFT and the corresponding DNS data. The essential details on the DNS methodology and properties of the series can be found in Sec. 2.2. Our approach was to repeat the analysis and to retrieve both basic characteristics of temperature profile and information on PSD at different cell's levels. Sample series can be seen in Fig. A1 and Fig. A2.
As presented in the Fig. 5 standard deviation distribution in the chamber is almost symmetrical with respect to the center with a small bias near the ceiling. The difference is attributed to the not perfectly insulated flange during UFT position adjustments. Fig. 10 present analogous study but with the DNS data covering exactly same thermodynamic conditions in the cell. Since the vertical grid size spans from about 1 mm near the plates to about 2.3 mm at the center, the available range significantly



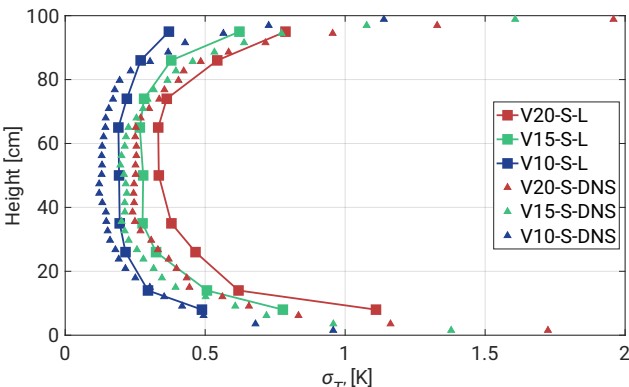

**Figure 10.** Standard deviation $\sigma_{T'}$ with respect to the height of the chamber. The chart analogical to Fig. 5 but including only 19 min segments (squares) and the DNS (triangles).

improves comprehensiveness of boundary layers. The limit regions exhibit maximum deviation of $\sigma_{T'} \approx 2\,\mathrm{K}$ with a tiny bias in
the vicinity of the upper plate. Also the shape of the curves is more bumped up in the center and slightly shifted left what might be an analogy to 3 min records in the Fig. 5. The numerical data provide more stable monotonicity but represents equivalent periods of time. In Fig. B1b there is non-dimensionalized form of this figure.

Similar conclusions can be made in terms of skewness profiles in Fig 11a. The DNS data exhibits much smaller fluctuations than corresponding 3 min UFT segments but preserve the general tendency near the floor and in the central region. The
characteristic jump in $\mu_{T'}$ is observed not around 20 cm but in half way. On the other side, a symmetrical jump is also observed near the ceiling what was not revealed in the UFT measurements likely due to a very shallow layer of thermal plume regime (the UFT measurements ended about 5 cm below the ceiling). Also the mixing region between 40–70 cm is reestablished resulting in higher deviations for lower $\Delta T$. The skewness distribution contributes to the mean vertical profile in the cell (see Fig. 11b). The presence of both positive (∼40 cm) and negative (∼80 cm) velocity jumps drives dynamics in the bulk region,
facilitating the mixing of cold and warm plumes. The horizontal components of the flow follow the LSC directions giving mean values of 15 cm/s near the plates. A more comprehensive discussion on the dynamics of the thermal plumes can be found in Sec. 3.1.

## 4   Summary

Our objective was to enhance understanding of thermally-driven convection in terms of small-scale variations in the temperature
scalar field. We conducted a small-scale study on the temperature structure of RBC in the Π Chamber using three different temperature differences (10 K, 15 K, and 20 K) at Ra of approximately $10^9$ and Pr of 0.7. Measurements were carried out using a miniaturized UltraFast Thermometer operating at 2 kHz, allowing undisturbed vertical temperature profiling from 8 cm above the floor to 5 cm below the ceiling. The key findings are summarized below.




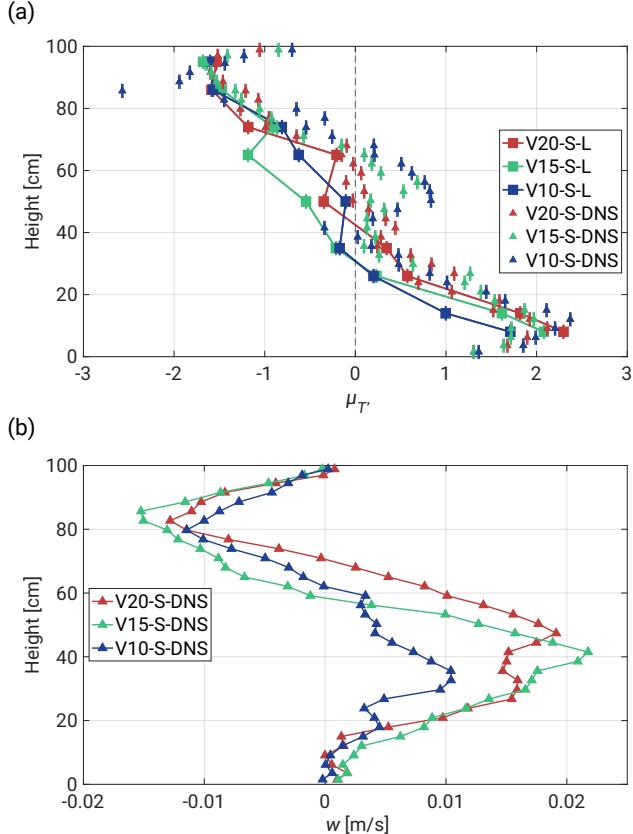

**Figure 11. (a)** Skewness $\mu_{T'}$ with respect to the height of the chamber. The chart analogical to Fig. 6 but including only 19 min segments (squares) and the DNS (triangles). **(b)** Mean vertical flow profile provided by the DNS.

– **basic characteristics**: We observed significant changes in the standard deviation and skewness of the distribution of
temperature fluctuations near the top and bottom surfaces. Additionally, we see variations in the scaling exponents of
the power spectrum in these near-surface regions. The turbulence in the center of the chamber exhibited characteristics
more akin to homogeneous, isotropic turbulence. These observed variations were attributed to the dynamics of local
thermal plumes and their interaction with the large-scale circulation (LSC). Both 19 min and 3 min measurements
were consistent, although the shorter records showed higher variability in standard deviation and skewness distribution.
The statistical properties of the temperature field obtained in RBC may offer insights into thermal processes in the
atmospheric surface layer, thereby enhancing our understanding of its thermal structure (Kukharets and Nalbandyan,
2006). Furthermore, the analysis of large-scale coherent structures in RBC provides a framework for understanding
thermal circulations, as well as the distribution of temperature and moisture, both in cloud chambers (Anderson et al.,
2021) and by analogy in the lower atmosphere (Zhou and Xia, 2013; Moller et al., 2021). The chamber is not designed



for idealized RBC experiments. Its structure solutions (e.g. windows on sides, atypical side-wall boundary conditions) aimed at cloud microphysics research is revealed in asymmetries of the profiles of temperature fluctuations statistics.

– **topographic effects**: No major differences were observed corresponding to topographic effects, likely due to insufficient time series. However, numerical work by Zhang et al. (2018) shed light on the necessary roughness height for robust heat transfer in RBC. Below the critical point, the authors observed trapped and accumulated heat inside the cavity regions
between the rough boundaries.

– **dynamic regimes**: PSD analysis revealed periodicity of LSC with respect to the temperature differences, characterized by an exponential formula consistent with previous findings (Anderson et al., 2021). We identified three distinct dynamic regimes: an inertial range (with slopes around $\sim -7/5$), a transition range (slopes around $\sim -3$), and a dissipative range (slopes around $\sim -7$). The scale break between the inertial and transition ranges was attributed to a dynamic
transition from the LSC-dominated regime to the thermal plume regime. Appendix D demonstrated that this transition is also observable in the spatial domain. Our findings are consistent with other studies not directly related to RBC though. For example, a similar scale break between inertial and transition ranges was observed in temperature fluctuation measurements near the surface of Jezero Crater on Mars (de la Torre Juárez et al., 2023), whereas corresponding slopes of $-17/3$ have been reported in power spectra of solar surface intensity variance field, attributed to buoyancy-driven
turbulent dynamics in a strongly thermally diffusive regime (Rieutord et al., 2010).

– **experiment versus DNS**: Experimental findings showed convincing agreement with DNS conducted under similar thermodynamic conditions, marking a rare comparative analysis in this field. Velocity profiles supported the argument for the nature of thermal plumes, and a method to convert spectra from the frequency domain to the wavenumber domain was detailed (see Appendix C). Despite the presence of imperfect boundaries such as window flanges, sampling ports, and
instrumentation, idealized DNS provided a reasonable representation of the actual Π Chamber flow, indicating that DNS adequately resolves surface layer fluxes. These results are valuable for improving and validating numerical research, such as sub-grid Large-Eddy Simulation models (Salesky et al., 2024), as well as global and local heat transport models (Scheel et al., 2013).

*Data availability.* The measurement records collected within this study are available from the authors upon request.

**Appendix A: Quicklooks of temperature fluctuations**

The figures illustrate the maximum local temperature fluctuations at the sensor's position. The presence of filaments or coherent structures, with temperatures close to that of the nearby plate, is clearly visible.



**Figure A1.** Experimental versus DNS $T'$ series ∼8 cm above the floor showed in the following time segments: 600 s, 60 s, 10 s, and 1 s. The used dataset covers $\Delta T = 20$ K case.





**Figure A2.** Experimental versus DNS $T'$ series ∼5 cm below the ceiling showed in the following time segments: 600 s, 60 s, 10 s, and 1 s. The used dataset covers $\Delta T = 20$ K case.





## Appendix B: Non-dimensional representation of standard deviations

Non-dimensionalized profiles of the standard deviation show stronger convergence in longer records and display greater sym-
metry compared to their dimensional representation.

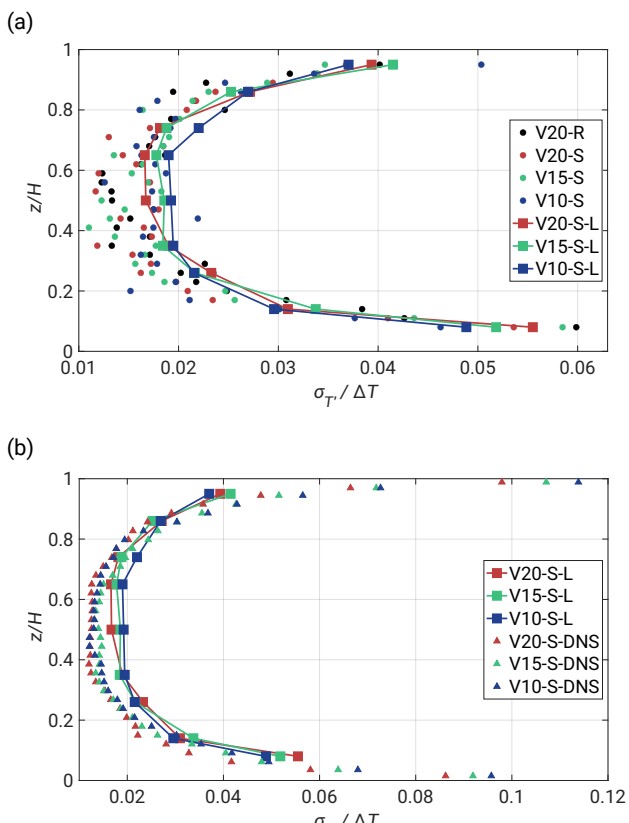

**Figure B1.** Non-dimensionalized standard deviation scaled by corresponding $\Delta T$. In the figure $z/H$ represents vertical distance $z$ measured from the bottom plate being normalized by the cell height $H = 1$ m. **(a)** Analogy of Fig. 5. **(b)** Analogy of Fig. 10.

## Appendix C: Power Spectral Density in wavenumber space

In the atmospheric community, PSD is typically presented in either the frequency or wavenumber domain, depending on preferences or scientific goals. As demonstrated in Section 3.2, the collapsed spectral curves in the frequency domain exhibit three dynamic regimes that characterize thermal convection in the Π Chamber. However, by following the scaling method

proposed by Kumar and Verma (2018) and the generalized approach of Zhou and Xia (2001), one can obtain analogous PSD in the wavenumber domain.

The first step of the scaling procedure involves the following transformations:



$$k \approx \tilde{f} = f(2\pi)/U,$$

$$P(k) \approx P(\tilde{f}) = P(f)U/2\pi.$$

where $P(\tilde{f})$ and $\tilde{f}$ represent the scaled frequency spectrum and scaled frequency, respectively. The DNS data revealed a
symmetrical profile of the mean velocity $U$ near the axis (see Fig. C1). It gradually decreases towards the bulk region, reaching about $0.02$ m/s, and maintains approximately equal values near both plates. The resulting wavenumber spectral curves are rescaled with $U$ and shifted accordingly. To collapse them, we found the Kolmogorov length scale, defined as $\eta = (2\pi)/k_n$, where $k_n$ is the wavenumber noise level, and performed another scaling to obtain the $P(\eta k)$ spectrum. The last step follows the adopted procedure of Zhou and Xia (2001).

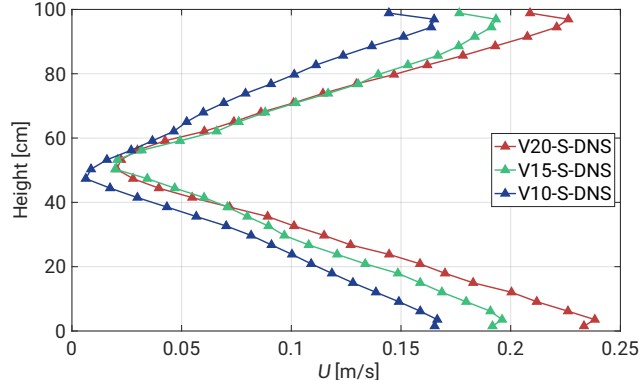

**Figure C1.** Mean velocity $U$ profile near the axis of the chamber with respect to its height. Each curve represent different $\Delta T$.

In Fig. C2a, we present the estimation of $k_\mathrm{p}\eta$, what is a direct analogy to $f_\mathrm{p}$ in Subsection 3.2, for the V20-S-L case in the scaled $k\eta^2 P(k\eta)$ spectrum. Unlike the corresponding plot in the frequency domain (see Fig. 8a), $k_\mathrm{p}\eta$ does not oscillate around one value. Here, we observe a gradual increase in $k_\mathrm{p}\eta$ values towards the bulk region, with the $k\eta^2 P(k\eta)$ maximum occurring around $k_\mathrm{p}\eta \approx 1/10$, which is in agreement with Monin and Yaglom (1975) (see Subsection 23.4 and Fig. 77). Fig. C2b provides the final result of the frequency to wavenumber scaling. Both the slopes and dynamic ranges are conserved, 395 providing a clear analogy to Fig. 8.

**Appendix D: Standard deviation scaling**

To verify if and where we can observe thermal plume dominance in the chamber, we followed the methodology outlined by He and Xia (2019). They demonstrated a strong connection between plumes and the logarithmic root mean square temperature profile using a different setup, which consisted of water as a working fluid ($\mathrm{Pr} = 4.34$), a rectangular-shaped container with 400 $\Gamma = 4.2$, and Ra varied from $3.2 \times 10^7$ to $2 \times 10^8$. For our purposes, we analyzed the standard deviation $\sigma_{T'}$ distribution of both the 19 min, 3 min, and DNS datasets. Fig. D1 displays the results for regions near both plates along with the respective





(a)

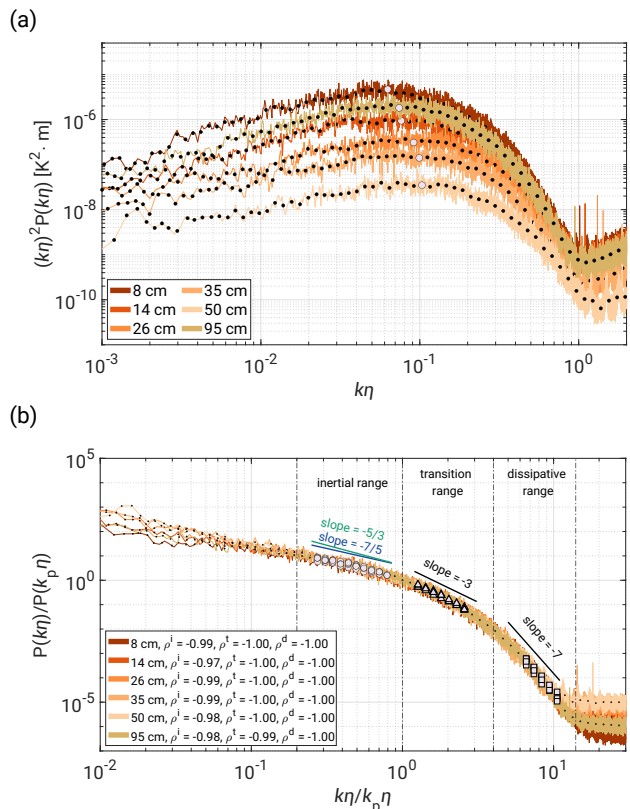

(b)

**Figure C2.** Analogical to Fig. 8 spectra of V20-S-L case but in wavenumber domain. **(a)** Scaled spectrum of $k\eta^2 P(k\eta)$. **(b)** PSD $P(k\eta)/P(k_{\mathrm{p}}\eta)$ versus $k\eta/k_{\mathrm{p}}\eta$ with three defined regimes: inertial range (circles, $0.2 \leqslant k\eta/k_{\mathrm{p}}\eta \leqslant 1$), transition range (triangles, $1 \leqslant k\eta/k_{\mathrm{p}}\eta \leqslant 4$), and dissipative range (squares, $4 \leqslant k\eta/k_{\mathrm{p}}\eta \leqslant 20$).

Pearson correlation coefficients in the semi-log domain. We selected regions approximately $\sim 15$–$35$ cm near the floor (Fig. D1a) and $\sim 65$–$95$ cm near the ceiling (Fig. D1b).

Each profile in the lower half of the chamber exhibits significant linearity correlation in the given region, including the 3 min experimental dataset. Only the V10-S case notably differs from the remaining results, dropping down to $p = -0.59$. The corresponding area in the upper half gives similarly high indications of the $p$ values, excluding shorter measurements, providing evidence of weaker thermal plume response. This observation is reasonable considering the previous discussion in Sec. 3.1 on differences between both regions of the cell.

It is worth mentioning that the zones outside the selected profiles are clearly dominated by different types of forces, resulting in very local dynamics in the RBC.



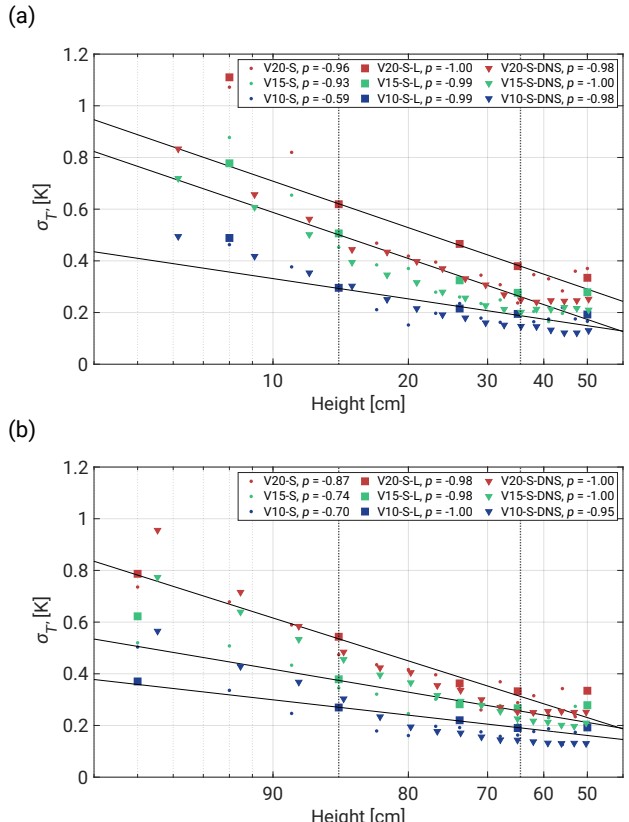

**Figure D1.** Standard deviation $\sigma_{T'}$ distribution of both 19 min, 3 min and DNS dataset in semi-log domain near the floor **(a)** and close to the ceiling **(b)**. Fitted curves correspond with 19 min measurements. Both legends include Pearson correlation coefficients.

*Author contributions.* RG, RAS, JCA, WC, and SPM designed the study. RG and JCA adapted the UFT instrument and the Π Chamber facility for the measurements. RG and JCA performed the UFT measurements. KKC performed DNS. RG processed and analysed the collected data with advice from RAS, SPM, and KKC. RG wrote the manuscript with contributions from RAS, SPM and KKC (who wrote Subsection 2.2). All authors critically proofread and revised the manuscript.

415    RG is truly grateful to Marta Wacławczyk and Jun-Ichi Yano for thought-provoking talks and for providing constructive feedback on our work.

*Competing interests.* The authors have the following competing interests: Szymon P. Malinowski is a member of the editorial board of Atmospheric Measurement Techniques.



*Acknowledgements.* This project has received funding from the Excellence Initiative-Research University Programme (IDUB) as part of
420    Action IV.4.1 under grant agreement No. BOB-IDUB-622-913/2023.

J. C. Anderson, W. Cantrell, and R. A. Shaw acknowledge support from the U.S. National Science Foundation through grant No. AGS-2113060.

K. K. Chandrakar's contribution was supported by the NSF National Center for Atmospheric Research, which is a major facility sponsored by the U.S. National Science Foundation under Cooperative Agreement No. 1852977.



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
