# Peer review of "High-resolution temperature profiling in the Π Chamber: variability of statistical properties of temperature fluctuations"

_EGUsphere, 2024_

## Referee Comment (RC1)

Review of:
**High-resolution temperature profiling in the Π Chamber:**
**variability of statistical properties of temperature fluctuations**
Manuscript: `egusphere-2024-2051`
**Authors:** R. Grosz, K.K. Chandrakar, R.A. Shaw, J.C. Anderson, W. Cantrell, and S.P. Malinowski

**Summary**

The authors present results from high-frequency (2 kHz) temperature measurements in the Michigan Technological University Pi chamber using an ultra-fast thermometer (UFT). They collect data at multiple heights throughout the depth of the chamber, and use the UFT data to analyze the temperature standard deviation and skewness and spectral properties of the temperature fluctuations. Overall this is an interesting manuscript and one that is well-suited for *Atmospheric Measurement Techniques*. However, I do have some comments I wish to have the authors address before publication, mainly dealing with clarity of presentation and some of the spectral analysis presented in the manuscript.

**General Comments**

1. **Review of previous work in RBC, l. 44 and following** I appreciate that the authors review the previous literature, but I found this section somewhat difficult to follow. Discussion of aspect ratio effects is interspersed with discussion of spectral scaling and Taylor's hypothesis. You may wish to consider revising this section so it is more clear to readers.

2. **ll. 185–200** Discussion of the role of the large-scale circulation and relationship to temperature skewness. I agree with this discussion in principle, but were these properties of the LSC measured in the present study? It just seems a bit more speculative. It may be a good idea to add some aadditional citations here with respect to the LSC and to make it clear to readers what was measured in the present study and which conclusions you are drawing based on previous work.

3. **l. 239 ff, discussion of spectral slopes** This is a very interesting part of the article, but I think the results get a bit buried in the discussion. Can the authors include a more explicit discussion of the predicted scalings for different regions of the spectrum from different sources? It's not completely clear from the text what the expect scalings are in different regions of the spectrum (or even whether predictions exist).

**Specific Comments**

1. **l. 82** What does $kS\,s^{-1}$ stand for here?

2. **l. 99 ff.** Discusssion of sampling period. "This variability stemmed from the LSC period ...and the uncertainty surrounding whether different turbulence properties might be observed for shorter time segments." I am not sure I understand from this passage why a variable measurement period was used. 3 minutes would only be about 2.5 large eddy turnover times. Is this enough to converge statistics?

3. **Fig. 4** "Top panel (a) corresponds to full vertical scan of the cell." I had to read the caption a couple of times before I understood that this single timeseries corresponds to different measurement heights. You may wish to revise so this is more clear to readers.

4. **l. 179** I'm not sure I like the notation $\mu'_T$ for the skewness. $\mu$ makes me think of the mean.

5. **l. 227 and Fig. 8** Why premultiply by $f^2$ (rather than $f$ or $f^{5/3}$ for instance)? This is not very clear from the text.

6. **l. 236 ff.** Discussion of BO and OC scaling. Have these acronyms been defined? I am not sure what you are referring to.

7. **Fig. 8 and discussion** Have the authors looked at the power spectral density pre-multiplied by $f$ to these different exponents? It may be interesting, for example to look at $f^{5/3}P(f)/P(f_p)$ or $f^{7/5}P(f)/P(f_p)$.

8. **l. 284** Spectral slope of $-7$ in dissipative range. Is it expected to have a power law in the dissipative range? The model spectra presented in Pope's textbook on *Turbulent Flows* (2000) includes exponential decay in the dissipative range. Granted this is for the energy spectrum rather than the scalar spectrum, but it may be beneficial to discuss this point further.

9. **Fig. 9, fitted slopes** Can the authors comment on how much uncertainty is present here in the fitted slopes? For momentum, if a fitted spectral slope did not correspond to $-5/3$ in the inertial range, I would suspect that the data are too noisy to estimate the slope accurately. Related to this, you may wish to include additional detail regarding how these spectral slopes were estimated.

10. **l. 330** Applicability of results to atmospheric surface layer. I don't disagree with this statement, but I will note that the atmospheric surface layer is typically shear-dominated, so there may not be a direct translation of the present results based on RBC data.

**Technical Corrections**

1. **l. 39** This is not a complete sentence.

2. **l. 56** Do you mean underlying, rather than underling?

3. **l. 277 ff.** This sentence does not read well; consider revising.

4. **Sec. 4, bullet point headings** These should be capitalized, e.g. "Basic Characteristics," "Topographic Effects," etc.

---

## Author Comment (AC2)

**Referee #2 responses**

(i) The manuscript has incorrect analysis and discussions on some of the data. In particular, some of the 3-minute measurement data is misinterpreted. It should be noted that this data is not statistically converged.

(ii) The conclusion and abstract need to indicate more of the limitations of the presented measurements.

(iii) Line 26 states some limitations, but this needs to be mentioned more in the data discussion to explain certain observations.

(iv) See a list of incorrect/inconsistent analyses and statements in the manuscript.

The manuscript presents results from both experimental and numerical analyses of high-resolution temperature series inside the Pi Chamber, with the former consisting of temperature time series measurements of 3 and 19 minutes. The chamber's primary objective is to investigate microphysical processes relevant to cloud formation development and decay, such as temperature, humidity and supersaturation fluctuations. Characteristic time of such processes is of the order of seconds and tens of seconds. The referee's primary concern is that the 3-minute data are too short to achieve convergence, as reflected in the skewness distribution in Fig. 6. We acknowledge this limitation; however, 3 - minute series provide valuable information on small-scale variability and inhomogeneity within the chamber useful for the microphysical applications, while  longer-term temperature properties useful for characterization of RBC inside the chamber are documented in the converged 19-minute series. The inclusion of the 3-minute data also serves to support our discussion on thermal plume dynamics and temperature variability relevant for atmospheric aerosol-cloud interaction experiments in the chamber. These short time series serve different purposes and complement the RBC analysis.

Concluding, we substantially modified the text of the manuscript,  adding relevant information to the abstract and discussion sections addressing the constraints of the 3-minute data.

-- Figure 3: Unclear what is learned from this figure.

The picture shows a simple schematic of our setup to visualise the used measurements methodology. It supports the description in the text.

-- Line 28: Suggest you do measurements in the boundary layer, which is incorrect. Boundary layer thickness is well below 1 cm for Ra=1e9, so the sentence should be removed.

Our intention was to indicate that the statistics we obtained in the Pi Chamber are comparable to the previous studies we provide references to. Those measurements were conducted in the boundary layer of RBC. For clarity we have improved this sentence.

-- Line 52: Fan et al not relevant

We disagree with the referee. The whole section III in this article discusses problems of RBC and discreetly heated surface which have direct links to our study, especially in terms of thermal plumes and atmospheric boundary layer connections. The paper also provides a broader perspective on natural convection which can provide better understanding of the problem for the readers loosely related with the RBC studies.

-- Line 55: Olsthoorn, 2023 not relevant

Context of line 55 was to show diversity of RBC problems in the literature, here shown from the theoretical side. It is also another example of translation of RBC consideration to geophysical context.

-- Line 71: Brown and Ahlers's most famous model on LSC is not cited, and not all given references are relevant

It has been included now in the manuscript, thank you for the suggestion.

Since no additional references have been explicitly identified as irrelevant, and from our perspective citations we included are relevant we cannot address this remark.

-- Line 77: See Annu. Rev. Fluid Mech. 2010. 42:335–64

Thank you for the suggestion, the paper is more focused on the structure functions analysis, however, we have added the citation for a better context of our study.

-- Line 143: Deteremination --> Determination

We have improved this typo.

-- Line 155-156: Why are no differences observed near the top plate? Roughness should still affect the flow.

In line 120 we state that the flange at the top was not ideally insulated from the outside conditions, additional comment is provided in line 160. This might be the main reason the roughness influence could have been suppressed.

-- Line 172: "All these effects are beyond the scope of this investigation, but the raw measurements give clear evidence of changing oscillations near both plates." --> This statement is incorrect. The fluctuations shown in Figure 4 cannot be compared to the LSC measured as performed in the RB community.

We have improved the sentence so that it does not confuse the reader. The meaning behind this part was to outline that temperature time series reveal some kind of irregular oscillations which might be attributed to LSC effects or other overlapping dynamic processes.

-- Scatter in these figures indicates that longer measurement data would be required. Performing measurements longer than 19 minutes would reduce fluctuations in statistical data.

We kindly draw the referee's attention that the main goal of the presented research is to explore small-scale variability of the scalar field which is of importance for microphysical processes in clouds, e.g. in terms of supersaturation fluctuations. The part on this starts in line 21 where we also included some citations of the recent works in a turbulent environment provided with the Pi Chamber facility.

-- Figure A1 and Figure A2: Temporal comparison between experiments and simulations is misleading as no one-to-one comparison of the same flow state is performed.

The intention here was to show two realizations (for each plate) indicating maximal fluctuations rather than one-to-one comparison. If we consider processes in the real atmosphere or in laboratory conditions, each particle in the flow experiences only local field fluctuations. These two overlapping curves exhibit quite similar magnitude of fluctuations between the DNS and our experiment despite the fact these two had different time resolutions. We have added an additional sentence to Appendix A to highlight the intention behind these plots.

-- line 195 and further: Discussion incorrect as the data are not converged. This data is not converged, and that discussion is meaningless. The 19-minute-long measurements do not show such an increase.

We agree with the referee that longer time series do not show the increase in skewness which is present in shorter records. It is put explicitly in line 193. However, we disagree with the statement that the following discussion is meaningless since it explores the potential reasons why we observe such change in skewness distribution near the lower plate. The timescale of thermal structures evolution is of seconds order which are averaged out in longer measurements. This is why even though 3 min data are not converged in the bulk region, we still can observe evidences of more organized behavior near the bottom. Moreover, the obtained results are also supported by DNS in Fig. 11 showing very similar skewness distribution.

-- line 211: Indeed 3, minute measurements are too short.

We have answered this point at the beginning of the document.

-- Figure 7: Vertical variability of the LSC period with respect to --> LSC is the coherent large scale flow structure in the cell, so it should not vertically vary. The discussion should be adjusted accordingly.

We agree with the referee and have improved the caption.

-- Provide details on how Pearson correlation coefficients are calculated.

The formula we used the Pearson correlation coefficient $p$ is the following,

$p(A, B) = \frac{1}{N-1} \sum_{i=1}^{N} = \left( \frac{A_i - \mu_A}{\sigma_A} \right) \left( \frac{B_i - \mu_B}{\sigma_B} \right)$, where e.g. for Fig. 8 coefficients $A$ and $B$ are

$\log\left(P(f)/p\left(f_p\right)\right)$ and $\log\left(f/f_p\right)$, $\mu$, $\sigma$, and $N$ are mean, standard deviation, and the number of observations respectively. For the best fit we also averaged the raw spectra over equidistant logarithmic frequency bins. The procedure is described in line 242.

-- Line 252: "relatively small variability of PSD slopes." --> The variations in the slopes are very large. Scaling exponents vary by as much as 20 to 30%. This severely limits how much one can learn from this data. That is recognized in line 254 ("Our analysis provides no clear answer,")

We have improved line 252. However, the slopes variation in the inertial regime increases as temperature difference decreases i.e. buoyant forces weaken. For

$\Delta T = 20$ K we observe slopes that exhibit smaller oscillations around -7/5 which then change near the top.

-- line 257-291: What main point do you want to explain to the reader?

In this part we investigate literature references to scalar slopes that would correspond to the results we obtained. This is why we explore potential connections with the other phenomena characteristics for 2D and 3D turbulence.
In the revised version we improved this paragraph and showed another link to thermal structures influence on scalar spectra resulting in the -3 regime which is based on a series of works by Chen and Bhaganaga (2021, 2023, 2024).
A comprehensive discussion of PSD scalar scaling in dissipative regime is provided by Gotoh and Yeung (2012) and Sreenivasan (2019) where it is indicated that for the Prandtl number of approximately 1 we do not know yet what the spectra in the roll-off region is. Some recent results for the energy spectrum have been provided by Khurshid et al. (2018) and Buaria and Sreenivasan (2020) based on direct interaction approximation by Kraichnan (1959), suggesting that this regime could be represented by an exponential form. However, here we show that power law might be sufficient for the scalar spectra in a buoyant environment.

-- line 339: It should be discussed that both measurement times are too short, especially the 3-minute data, which even leads to misleading discussions in the manuscript.

We have answered this point at the beginning of the document.

-- line 341: How this improves our understanding of atmospheric flow is not discussed.

This part has been improved. The meaning behind this was to point out that RBC experiments on thermal structures, like thermal plumes, might be very valuable in terms of better understanding of the nature of surface-air temperature fluctuations through statistical analyses.

-- line 368: Scheel et al. is a numerical study with no theory on heat transport.

We have improved the citation.

-- Figure 11: Why is DNS not symmetric around the mid-plane?

The plot represents single-column data (not the horizontal average) this is why the perfect symmetry in mean $w$ and $T$ skewness is not expected, in particular for the period of LSC circulation. The appropriate note has been added in the figure description.

-- Appendix B: "and display greater symmetry compared" --> Incorrect. Normalising the temperature difference between the plates does not affect symmetry around the midplane.

We agree and have improved this part.

-- Figure C1: What does it mean that velocity in the centre is always positive? Is this a velocity magnitude?

Yes, this is the magnitude of the mean velocity profile. The caption has been updated.

-- Figure C2: What are all the numbers in the legend?

In C2a numbers describe the positions of the UFT with respect to the chamber's height. In C2b these are the Pearson coefficients. Both graphs are a direct analogy of Fig. 8 but in wavenumber space.

-- Figure D1: The data are visibly deviating from the presented lines. Difficult to see how this corresponds to Pearson correlation coefficients of -1.

The Pearson coefficients are calculated for all three sets of data (3 min, 19 min, and the DNS) but the lines are fitted only for 19 min datasets (squares). We have added the missing information on this in the manuscript.

---

## Author Response (AR1)

**Referee #1 reponses**

**General Comments**

1. **Review of previous work in RBC, l. 44 and following** I appreciate that the authors review the previous literature, but I found this section somewhat difficult to follow. Discussion of aspect ratio effects is interspersed with discussion of spectral scaling and Taylor's hypothesis. You may wish to consider revising this section so it is more clear to readers.

Following the advice, we reorganized this part to be more consistent and not to confuse the readers. The modified section starts at line 52.

2. **ll. 185–200** Discussion of the role of the large-scale circulation and relationship to temperature skewness. I agree with this discussion in principle, but were these properties of the LSC measured in the present study? It just seems a bit more speculative. It may be a good idea to add some additional citations here with respect to the LSC and to make it clear to readers what was measured in the present study and which conclusions you are drawing based on previous work.

Our measurements of the LSC focused on the oscillation periods as a function of the applied temperature differences. The resulting power law aligns with previously measured periods for $T = 12\,°\text{C}$. Details regarding these measurements are provided in the abstract and in Section 3.2, where we present the PSD results. The subsequent discussion on skewness properties represents our attempt to propose a potential mechanism responsible for the observed effects. This section has been refined to improve clarity, and we have incorporated citations to relevant works to better contextualize our findings. The modified section starts at line 208.

3. **l. 239 ff**, discussion of spectral slopes This is a very interesting part of the article, but I think the results get a bit buried in the discussion. Can the authors include a more explicit discussion of the predicted scalings for different regions of the spectrum from different sources? It's not completely clear from the text what the expect scalings are in different regions of the spectrum (or even whether predictions exist).

We revised this part providing the information of slopes predictions, clarifying the discussion of our results and improving the argumentation based on other works. The modified section starts at line 259.

**Specific Comments**

1. **l. 82** What does kS s$^{-1}$ stand for here?

It stands for kilosamples per second. This is explained now in the text. The modified part starts at line 99.

2. **l. 99 ff.** Discusssion of sampling period. "This variability stemmed from the LSC period. . . and the uncertainty surrounding whether different turbulence properties might be observed for shorter time segments." I am not sure I understand from this passage why a variable measurement period was used. 3 minutes would only be about 2.5 large eddy turnover times. Is this enough to converge statistics?

Our objective was to investigate the small-scale variability of the scalar field, which plays a critical role in microphysical processes within clouds. The use of variable measurement periods was motivated by several factors: the need to capture scalar profiles with fine spatial resolution to better understand the physics at different levels of the chamber, the high temporal resolution of the measurements (2 kHz), which provided robust statistics even for shorter durations, and the limited availability of the Pi Chamber, which constrained the collection of longer time series.

Figure 5 demonstrates that both longer and shorter time series exhibit similar overall behavior. However, as shown in Fig. 6 and discussed subsequently, shorter time series reveal probable unaveraged thermal structures that are not apparent in longer measurements, emphasizing features that would otherwise be smoothed out.

In the revised manuscript, we included a paragraph providing a quantitative estimate of data convergence based on the framework of Lenschow et al. (1994). Specifically, we estimated that, in our measurements, the large eddy correlation time corresponds to the turbulence flow correlation time, which in the Pi Chamber is on the order of a few seconds. Consequently, both time series demonstrate good convergence. A more detailed explanation is provided starting in line 116 of the revised manuscript.

3. **Fig. 4** "Top panel (a) corresponds to full vertical scan of the cell." I had to read the caption a couple of times before I understood that this single timeseries corresponds to different measurement heights. You may wish to revise so this is more clear to readers.

We revised the caption to improve readability.

4. **l. 179** I'm not sure I like the notation $\mu'_T$ for the skewness. $\mu$ makes me think of the mean.

We changed the notation ($\mu \rightarrow \gamma$)

5. **l. 227** and Fig. 8 Why premultiply by $f^2$ (rather than $f$ or $f^{5/3}$ for instance)? This is not very clear from the text.

We explained the methodology used starting from line 235, noting that it was originally proposed by Zhou and Xia (2001). The cited paper provides a more comprehensive discussion of this approach. Using this method, we were able to identify the peak frequencies around which the PSDs converge to universal functions. In the revised manuscript this section starts at line 259.

6. **l. 236 ff.** Discussion of BO and OC scaling. Have these acronyms been defined? I am not sure what you are referring to.

In line 76 there was an explanation of the used abbreviations. Both denote different scalings—OC stands for Obukhov-Corrsin whereas BO is Bolgiano-Obukhov. In the revised manuscript this part is explained in line 78.

7. **Fig. 8 and discussion** Have the authors looked at the power spectral density pre-multiplied by f to these different exponents? It may be interesting, for example to look at $f^{5/3}P(f)/P(f_p)$ or $f^{7/5}P(f)/P(f_p)$.

The normalized spectra mentioned are presented below, corresponding to the case of $\Delta T = 20$ K, as described in the manuscript. The graphs show results consistent with the prior discussion: the inertial range is slightly better characterized by the −7/5 scaling. However, due to the minimal difference between −5/3 and −7/5, it is challenging to definitively conclude which scaling is more representative. The root mean square errors for this range are addressed in point 9.

[Figure]

8. l. 284 Spectral slope of -7 in dissipative range. Is it expected to have a power law in the dissipative range? The model spectra presented in Pope's textbook on *Turbulent Flows* (2000) includes exponential decay in the dissipative range. Granted this is for the energy spectrum rather than the scalar spectrum, but it may be beneficial to discuss this point further.

A comprehensive discussion on scalar spectra with respect to Schmidt number (Sc) can be found in Sreenivasan, K. R., *Proceedings of the National Academy of Sciences*, **116**(37), 18175–18183, doi:10.1073/pnas.1800463115 (2019). For the case of Sc ≈ 1, what can be translated to magnitude on the order of unity of the Prandtl number, there is no theory determining scalar field in the dissipative range. Since temperature acts as active scalar the resulting spectra can differ from the respected energy spectrum. In case of spectra we obtained, the slopes were estimated as power law functions which fitted well. More on this we included in the next point as well as in the extended discussion in the manuscript. In the revised manuscript the corresponding discussion starts at line 266.

9. **Fig. 9, fitted slopes** Can the authors comment on how much uncertainty is present here in the fitted slopes? For momentum, if a fitted spectral slope did not correspond to -5/3 in the inertial range, I would suspect that the data are too noisy to estimate the slope accurately. Related to this, you may wish to include additional detail regarding how these spectral slopes were estimated.

The methodology of slopes estimation was outlined in the manuscript starting from the line 241: "To estimate the slopes, we employed a methodology outlined in Siebert et al. (2006) and Nowak et al. (2021), averaging raw spectra over equidistant logarithmic frequency bins (twenty bins per decade in our case) and fitting using power law functions. Additionally, to assess the linearity of the slopes in log-log coordinates, we computed the Pearson correlation coefficient $p$ for the resampled points."

[Figure]

[Figure]

(e)

The equidistant logarithmic bins are presented as black points in Fig. 8, then we selected segments of points (marked as circles, triangles and squares in Fig. 8b) with the highest Pearson coefficients for the best power law fits. The resulting slopes are then presented in Fig. 9 (we attached both panels as (a) and (b) graphs above). To show the uncertainties of the fitted slopes we exported root mean squared errors (RMSE) in graphs above: inertial range (c), transition range (d), and dissipative range (e). The largest RMSE are indeed observable for the inertial range which is why we cannot make a definite conclusion of which scaling is better represented in our results. The following ranges are characterized by small (d) and very small (e) uncertainties which support the idea of power law dependence in dissipative range for scalar field.

10. **l. 330** Applicability of results to atmospheric surface layer. I don't disagree with this statement, but I will note that the atmospheric surface layer is typically shear dominated, so there may not be a direct translation of the present results based on RBC data.

We agree with the referee that the atmospheric boundary layer represents very complex physics and shear effects are significant. This is why we merely outlined the possible connections between the real atmosphere and our results as thermal structures play an important role in mixing processes. Nevertheless, we think that application of the UFT thermometers in the atmospheric surface layer could give a lot of interesting information about its thermal structure.

**Technical Corrections**

1. **l. 39** This is not a complete sentence.

The sentence was improved.

2. **l. 56** Do you mean underlying, rather than underling?

Unfortunately, we do not find neither a word "underling" nor "underlying" in our work.

3. **l. 277 ff.** This sentence does not read well; consider revising.

The sentence was improved.

4. **Sec. 4, bullet point headings** These should be capitalized, e.g. "Basic Characteristics,"
"Topographic Effects," etc.

We improved the bullet point headings following the referee's advice.

**Referee #2 responses**

(i) The manuscript has incorrect analysis and discussions on some of the data. In particular, some of the 3-minute measurement data is misinterpreted. It should be noted that this data is not statistically converged.

(ii) The conclusion and abstract need to indicate more of the limitations of the presented measurements.

(iii) Line 26 states some limitations, but this needs to be mentioned more in the data discussion to explain certain observations.

(iv) See a list of incorrect/inconsistent analyses and statements in the manuscript.

The manuscript presents results from both experimental and numerical analyses of high-resolution temperature series inside the Pi Chamber, with the former consisting of temperature time series measurements of 3 and 19 minutes. The chamber's primary objective is to investigate microphysical processes relevant to cloud formation development and decay, such as temperature, humidity and supersaturation fluctuations. Characteristic time of such processes is of the order of seconds and tens of seconds. The referee's primary concern is that the 3-minute data are too short to achieve convergence, as reflected in the skewness distribution in Fig. 6. In the revised manuscript, we included a paragraph providing a quantitative estimate of data convergence based on the framework of Lenschow et al. (1994). Specifically, we estimated that, in our measurements, the large eddy correlation time corresponds to the turbulence flow correlation time, which in the Pi Chamber is on the order of a few

seconds. Consequently, both time series demonstrate good convergence. A more detailed explanation is provided in line 116 in the revised manuscript.

-- Figure 3: Unclear what is learned from this figure.

The picture shows a simple schematic of our setup to visualise the used measurements methodology. It supports the description in the text.

-- Line 28: Suggest you do measurements in the boundary layer, which is incorrect. Boundary layer thickness is well below 1 cm for Ra=1e9, so the sentence should be removed.

Our intention was to indicate that the statistics we obtained in the Pi Chamber are comparable to the previous studies we provide references to. Those measurements were conducted in the boundary layer of RBC. For clarity we have improved this sentence. In the revised manuscript this part starts in line 28.

-- Line 52: Fan et al not relevant

We disagree with the referee. The whole section III in this article discusses problems of RBC and discreetly heated surface which have direct links to our study, especially in terms of thermal plumes and atmospheric boundary layer connections. The paper also provides a broader perspective on natural convection which can provide better understanding of the problem for the readers loosely related with the RBC studies.

-- Line 55: Olsthoorn, 2023 not relevant

Context of line 55 was to show diversity of RBC problems in the literature, here shown from the theoretical side. It is also another example of translation of RBC consideration to geophysical context.

-- Line 71: Brown and Ahlers's most famous model on LSC is not cited, and not all given references are relevant

It has been included now in the manuscript, thank you for the suggestion.

Since no additional references have been explicitly identified as irrelevant, and from our perspective citations we included are relevant we cannot address this remark. In the revised manuscript the citation is included in line 70.

-- Line 77: See Annu. Rev. Fluid Mech. 2010. 42:335−64

Thank you for the suggestion, the paper is more focused on the structure functions analysis, however, we have added the citation for a better context of our study.

-- Line 143: Deteremination --> Determination

We have improved this typo.

-- Line 155-156: Why are no differences observed near the top plate? Roughness should still affect the flow.

In line 120 we stated that the flange at the top was not ideally insulated from the outside conditions, additional comment was provided in line 160. This might be the main reason the roughness influence could have been suppressed.

-- Line 172: "All these effects are beyond the scope of this investigation, but the raw measurements give clear evidence of changing oscillations near both plates." --> This statement is incorrect. The fluctuations shown in Figure 4 cannot be compared to the LSC measured as performed in the RB community.

We have improved the sentence so that it does not confuse the reader. The meaning behind this part was to outline that temperature time series reveal some kind of irregular oscillations which might be attributed to LSC effects or other overlapping dynamic processes. In the revised manuscript the modified part starts in line 184.

-- Scatter in these figures indicates that longer measurement data would be required. Performing measurements longer than 19 minutes would reduce fluctuations in statistical data.

We kindly draw the referee's attention that the main goal of the presented research is to explore small-scale variability of the scalar field which is of importance for microphysical processes in clouds, e.g. in terms of supersaturation fluctuations. The part on this started in line 21 where we also included some citations of the recent works in a turbulent environment provided with the Pi Chamber facility.

-- Figure A1 and Figure A2: Temporal comparison between experiments and simulations is misleading as no one-to-one comparison of the same flow state is performed.

The intention here was to show two realizations (for each plate) indicating maximal fluctuations rather than one-to-one comparison. If we consider processes in the real atmosphere or in laboratory conditions, each particle in the flow experiences only local field fluctuations. These two overlapping curves exhibit quite similar magnitude of fluctuations between the DNS and our experiment despite the fact these two had different time resolutions. We have added an additional sentence to Appendix A to highlight the intention behind these plots.

-- line 195 and further: Discussion incorrect as the data are not converged. This data is not converged, and that discussion is meaningless. The 19-minute-long measurements do not show such an increase.

We agree with the referee that longer time series do not show the increase in skewness which is present in shorter records. It was put explicitly in line 193. However, we disagree with the statement that the following discussion is meaningless since it explores the potential reasons why we observe such change in skewness distribution near the lower plate. The timescale of thermal structures evolution is of seconds order which are averaged out in longer measurements. Moreover, the obtained results are also supported by DNS in Fig. 11 showing very similar skewness distribution. We provide a discussion on statistical data convergence in line 116 in the revised manuscript.

-- line 211: Indeed 3, minute measurements are too short.

We have answered this point at the beginning of the responses.

-- Figure 7: Vertical variability of the LSC period with respect to --> LSC is the coherent large scale flow structure in the cell, so it should not vertically vary. The discussion should be adjusted accordingly.

We agree with the referee and have improved the caption.

-- Provide details on how Pearson correlation coefficients are calculated.

The formula we used the Pearson correlation coefficient $p$ is the following, $p(A, B) = \frac{1}{N-1} \sum_{i=1}^{N} \left(\frac{A_i - \mu_A}{\sigma_A}\right)\left(\frac{B_i - \mu_B}{\sigma_B}\right)$, where e.g. for Fig. 8 coefficients $A$ and $B$ are $\log\left(P(f)/p(f_p)\right)$ and $\log(f/f_p)$, $\mu$, $\sigma$, and $N$ are mean, standard deviation, and the number of observations respectively. For the best fit we also averaged the raw spectra over equidistant logarithmic frequency bins. The procedure is described in line 262 in the revised manusript.

-- Line 252: "relatively small variability of PSD slopes." --> The variations in the slopes are very large. Scaling exponents vary by as much as 20 to 30%. This severely limits how much one can learn from this data. That is recognized in line 254 ("Our analysis provides no clear answer,")

We have improved this section. However, the slopes variation in the inertial regime increases as temperature difference decreases i.e. buoyant forces weaken. For $\Delta T = 20$ K we observe slopes that exhibit smaller oscillations around -7/5 which then change near the top. The updated section starts at line 271 in the revised manuscript.

-- line 257-291: What main point do you want to explain to the reader?

In this part we investigate literature references to scalar slopes that would correspond to the results we obtained. This is why we explore potential connections with the other phenomena characteristics for 2D and 3D turbulence.

In the revised version we improved this paragraph and showed another link to thermal structures influence on scalar spectra resulting in the -3 regime which is based on a series of works by Chen and Bhaganaga (2021, 2023, 2024).

A comprehensive discussion of PSD scalar scaling in dissipative regime is provided by Gotoh and Yeung (2012) and Sreenivasan (2019) where it is indicated that for the Prandtl number of approximately 1 we do not know yet what the spectra in the roll-off region is. Some recent results for the energy spectrum have been provided by Khurshid et al. (2018) and Buaria and Sreenivasan (2020) based on direct interaction approximation by Kraichnan (1959), suggesting that this regime could be represented by an exponential form. However, here we show that power law might be sufficient for the scalar spectra in a buoyant environment. The discussion starts at line 282 in the revised manuscript.

-- line 339: It should be discussed that both measurement times are too short, especially the 3-minute data, which even leads to misleading discussions in the manuscript.

We have answered this point at the beginning of the document.

-- line 341: How this improves our understanding of atmospheric flow is not discussed.

This part has been improved. The meaning behind this was to point out that RBC experiments on thermal structures, like thermal plumes, might be very valuable in terms of better understanding of the nature of surface-air temperature fluctuations through statistical analyses. The updated summary starts at line 356 in the revised manuscript.

-- line 368: Scheel et al. is a numerical study with no theory on heat transport.

We have improved the citation. It can be found in line 402 in the revised manuscript.

-- Figure 11: Why is DNS not symmetric around the mid-plane?

The plot represents single-column data (not the horizontal average) this is why the perfect symmetry in mean $w$ and $T$ skewness is not expected, in particular for the period of LSC circulation. The appropriate note has been added in the figure description.

-- Appendix B: "and display greater symmetry compared" --> Incorrect. Normalising the temperature difference between the plates does not affect symmetry around the midplane.

We agree and have improved this part.

-- Figure C1: What does it mean that velocity in the centre is always positive? Is this a velocity magnitude?

Yes, this is the magnitude of the mean velocity profile. The caption has been updated.

In C2a numbers describe the positions of the UFT with respect to the chamber's height. In C2b these are the Pearson coefficients. Both graphs are a direct analogy of Fig. 8 but in wavenumber space.

-- Figure D1: The data are visibly deviating from the presented lines. Difficult to see how this corresponds to Pearson correlation coefficients of -1.

The Pearson coefficients are calculated for all three sets of data (3 min, 19 min, and the DNS) but the lines are fitted only for 19 min datasets (squares). We have added the missing information on this in the manuscript.

---

## Author Response (AR2)

**Authors' statement**

We would like to inform you that, during a careful review of the manuscript and codes, we identified a minor miscalculation in one of the scripts, for which we sincerely apologize. This results in slight modifications to the plots in Fig. C2 in Appendix C (visible changes in chart (a)) and the rewording of two sentences—one in Appendix C and another in the spectral discussion in Subsection 3.2. These adjustments do not affect the interpretation of the results.

**Review reponses**

If referring to a specific line in the manuscript, we consider the revised version.

**Referee #1**

All the technical corrections have been incorporated in the revised version. Thank you very much for the suggestions and the review.

**Referee #2**

The presented study does not focus on the canonical scientific problems typically addressed in the RB community. The research had an objective to thermally characterize the chamber in terms of small-scale events. It is essential for future experiments in the Pi Chamber, providing insights into temporal scales and the magnitude of small-scale scalar variability, as well as the periodicity of the LSC under different temperature differences.

Processes occurring in the real atmosphere are neither stationary nor stable and exhibit significant variability, making them particularly challenging to study, especially at turbulence scales. Additional analysis of the results along the lines of canonical studies of RBC, despite differing convergence criteria from those traditionally recognized in the RBC community, allows to understand conditions inside the Pi-chamber in a broader and more general context. Nonetheless, the 10% range ensures satisfactory convergence for atmospheric applications. We have revised the abstract and introduction to better emphasize the paper's objective.

Additionally, in response to the reviewer's concerns, we clarify that the choice of 3- and 19-minute measurement durations was dictated by the Pi Chamber's operational schedule, which constrained both the number of sampled points and the range of explored conditions.

Please find below some remaining comments.

homogeneous horizontal grid spacing --> Does this mean boundary layers closer to the vertical sidewall are not explicitly resolved?

A uniform horizontal grid spacing of 2.083 mm is used for DNS. This grid spacing is lower than the thickness of the top and bottom boundary layer (~ 5 mm). The sidewall heat flux is significantly smaller than the top and bottom surfaces for isothermal sidewalls. Therefore, the horizontal resolution should sufficiently capture the sidewall boundary layer.

How many points are in the boundary layers close to the plates?

The positions of DNS grid points translated into spatial units are as follows (considering only 20 cm apart from each plate): 1.5; 3.5; 6.2; 9.1; 12.0; 15.0; 17.9; and 82.7; 85.6; 88.6; 91.5; 94.5; 96.9; 98.9 [cm].

On what is the statement that previous studies had "limited temporal resolution" based?

"…limited temporal resolution of the used instrumentation." statement refers to temporal resolution of instrumentation used in the reported RBC studies which could not have resolved full range of scales especially the smallest scales of turbulence.

"As presented in Fig. 5, standard deviation distribution in the chamber is almost symmetrical with respect to the center with a small bias near the ceiling." --> No; this does not reflect the discussion on the asymmetry as indicated in my previous report and the answer to the referee. Please update the discussion as indicated in the response to the referee

Thank you for drawing our attention to this issue. It is now incorporated in the discussion. The same applies to Appendix A.

Moreover, we rephrased some sentences to avoid being inaccurate.

**Referee #3**

1. The temperature fluctuation profiles in Fig. 5 (and Fig. 11) look meaningful to me. The authors cannot resolve the near-wall regions such that they miss the profile maximum at the height of the thermal boundary layer thickness. Shouldn't the third-order moment in Fig. 6 be more or less mirror-symmetric (about half-height)? It might be that 19 min of measurement time are still too short. Could the authors comment on

this. For example, how much does the LSC change in this time period, can this be detected in the experiment?

During the experiments, we used two UFT head sensors, which could not be easily repaired in the event of damage. To mitigate this risk, we intentionally limited the vertical measurement range.

Ideally, we would expect a more symmetrical skewness distribution. However, the chamber does not represent a perfect example of RBC, partly due to the top insulation design during the experiment. Additionally, we manually adjusted the sensor position each time, which required removing the foam lid at the top for approximately two minutes. The procedure, described in detail starting from line 133, likely allowed some ambient air from the laboratory to enter the chamber. This could have introduced fluctuations in the third statistical moment, which is highly sensitive to even minor disturbances.

According to Fig. 7, the LSC oscillation period varies from about 1.3 min (at 10 K) to approximately 1 min (at 20 K), corresponding to roughly 14 and 19 full cycles, respectively. Since our measurements covered only the vertical column near the axis, assessing LSC stability during that period remains challenging. We refer to relevant literature discussing this issue as a more complex phenomenon (starting from line 68).

2. On the spectra: a Bolgiano scaling should appear for scales above the Bolgiano scale L_B (which might be translated into a Bolgiano frequency f_B = u_rms L_B. Is it possible to extract such a scale (e.g. from the DNS) that separates the lowest frequencies from the rest of the spectrum? Otherwise, one might conclude that it is simply a Kolmogorov-type scaling. Bolgiano scaling is typically found in stratified turbulence, not in unstably stratified flows.

An extended discussion of the limitations of BO scaling in RBC is provided by Lohse and Xia (2010). The most significant point is that in canonical RBC, the difficulties in direct scale separation suggests that BO scaling may be limited. However, the authors in Fig. 5 showed that BO scaling has been reported in such system previously.

Moreover, Lohse and Xia discussed that Bolgiano length should be estimated locally based on energy and temperature dissipation rates. In our study, we examined scaling regimes based purely on the linearity condition (Pearson correlation coefficient), showing that within the inertial range, the slopes exhibit a slight bias toward −7/5 (particularly at 20 K). However, we also noted that the slopes of −5/3 and −7/5 are too similar to draw a definitive conclusion. Additionally, we considered the possibility of a gradual spectral transition rather than a sharp distinction between scaling regimes.

3. The discussion of the -3 range of the spectrum: can we really see fingerprints of a helicity cascade in the spectrum. The spectra are taken along (or close to) the centerline. Perhaps it just might be a crossover into the dissipative range. Please comment.

We interpret that our scaling observations in this regime are linked to the dynamics of thermal plumes according to the works of Chen and Bhaganagar. In their series of studies, they argued that the $-5/3$ and $-3$ scalings could be associated with helicity cascades. While the exact nature of this invariant remains unclear, as noted in our manuscript, it presents an intriguing direction for future research. At this stage, however, it is challenging to provide a definitive explanation.

Thank you for the suggestion though, we incorporated information on this possibility in the revised version of the manuscript.

---

## Author Response (AR3)

line 30/31: "This is because phenomena present in real atmosphere undergo non-stationary and unstable processes which are difficult to study in natural conditions." Did you mean "... are difficult to study in laboratory conditions."?

In this part we refer to challenges that scientists face when trying to study atmospheric phenomena in natural conditions (real atmosphere). To improve clarity, we have revised the sentence as follows: *Atmospheric phenomena undergo non-stationary and unstable processes, making them difficult to study in real atmosphere.*

line 114 and figure 3, caption: It is not clear, 1) what, 2) where, and 3) for what the cylinder is that you refer to.

We have incorporated necessary improvements to provide a more informative description of the experiments.

3/19 min measurement times: You provided information on why you chose these times in the review replies, but not in the manuscript, Please include a statement into the manuscript as well. (e.g around line 121)

We have included the requested information in the manuscript.

Figure 8. "Scaled V20-S-L PSD with respect to the UFT positions (color gradients) in the lower half of the chamber (plus top position)." The label in Fig. 8a does suggest positions throughout the chamber?

The top panel covers all V20-S-L positions across the chamber volume, while the lower panel includes only measurements from the lower half of the chamber and the top position. We have rephrased the caption for clarity.

Figure A1 and A2: Please also note in the figure caption that the time series of measurement and DNS simulation do not correspond to each other.

The requested notes have now been incorporated into both captions.